# ERROR NOTEBOOK-GUIDED, TRAINING-FREE PART RETRIEVAL IN 3D CAD ASSEMBLIES VIA VISION-LANGUAGE MODELS

**Yunqing Liu, Nan Zhang & Zhiming Tan**[*]
Fujitsu Research & Development Center
Shanghai, China
`{liuyunqing, zhangnan, zhmtan}@fujitsu.com`

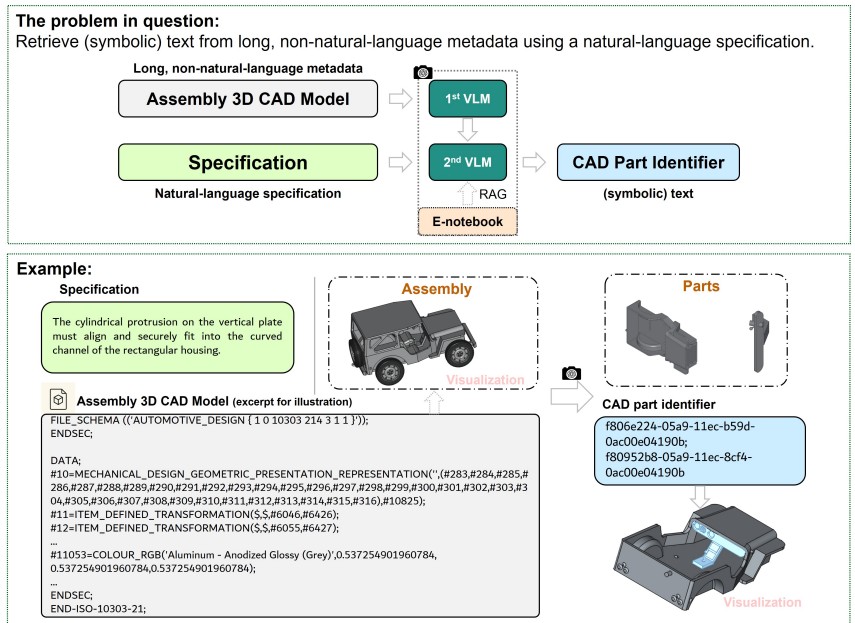

Figure 1: **Scope of work.** The goal is to retrieve symbolic part identifiers from long, non-natural-language assembly CAD model using a natural-language specification. Our two-stage VLM pipeline first converts CAD model part information into geometric descriptions (1st VLM), then performs specification-aware reasoning (2nd VLM) assisted by *Error Notebook* with RAG.

## ABSTRACT

Effective **specification-aware part retrieval** within complex CAD assemblies is essential for automated engineering tasks. However, using LLMs/VLMs for this task is challenging: the CAD model metadata sequences often exceed token budgets, and fine-tuning high-performing proprietary models (e.g., GPT or Gemini) is unavailable. Therefore, we need a framework that delivers engineering value by handling long, non-natural-language CAD model metadata using VLMs, but without training. We propose a 2-stage framework with **inference-time adaptation** that combines corrected *Error Notebooks* **with RAG** to substantially improve VLM-based part retrieval reasoning. Each *Error Notebook* is built by correcting initial CoTs through reflective refinement, and then filtering each trajectory using our proposed grammar-constraint (GC) verifier to ensure structural well-formedness. The resulting notebook forms a high-quality repository of specification-CoT-answer triplets, from which RAG retrieves specification-relevant exemplars to condition the model's inference. We additionally contribute

---

[*]Corresponding author.

a CAD dataset with **human preference annotations**. Experiments with proprietary models (GPT-4o, Gemini, etc) show large gains, with GPT-4o (Omni) achieving up to +23.4 absolute accuracy points on the human-preference benchmark. The proposed GC verifier can further produce up to +4.5 accuracy points. **Our approach also surpasses other training-free baselines (standard few-shot learning, self-consistency) and yields substantial improvements also for open-source VLMs (Qwen2-VL-2B-Instruct, Aya-Vision-8B). Under the cross-model GC setting, where the *Error Notebook* is constructed using GPT-4o (Omni), the 2B model inference achieves performance that comes within roughly 4 points of GPT-4o mini.**

# 1 INTRODUCTION

Recent efforts demonstrate the promise of LLMs/VLMs in the engineering design and manufacturing domain. For instance, Alrashedy et al. (2025) applied LLMs to generate Computer-Aided Design (CAD) code from natural language descriptions, which can then be executed to render 3D objects. Such approaches show that general-purpose models can automate the CAD modeling process. Additionally, Wu et al. (2021) developed deep generative models to create 3D CAD structures directly (e.g., by modeling sequences of CAD operations), hinting at the potential of combining language and vision for CAD design tasks. Recent work has also shown that LLMs can assist in design ideation and automation, such as guiding parametric modeling, generating shape descriptions, and integrating CAD workflows with natural language instructions (Li et al. (2025); Zhang et al. (2025); Kumar et al. (2025)). These studies further highlight the versatility of LLMs in supporting creative and engineering tasks within CAD environments. Despite this progress, a critical task remains challenging for LLMs/VLMs: specification-driven part retrieval within complex CAD assemblies. Each CAD assembly (e.g., stored as a STEP file) can contain dozens of parts described by lengthy, non-natural language metadata. Retrieving specific parts that match a given design specification or relational description is essential for automated design verification and other downstream tasks, yet directly prompting LLMs or VLMs for this often yields poor results. A primary obstacle is the extreme sequence length of assembly data, which can exceed current model token limits. Even if the STEP data is processed, for example, into images, we found that off-the-shelf models still frequently misidentify parts because the task requires fine-grained reasoning about part relationships and attributes.

Fine-tuning a model on this task could improve performance, but it is sometimes impractical: many models (e.g., GPT or Gemini) are proprietary or lack fine-tuning access, and training a custom model would demand significant computational resources. However, certain training techniques for LLMs and VLMs may serve as inspiration for enhancing the performance of methods that do not require training. For example, in the mathematical domain, Pan et al. (2025) fine-tuned a model on a special dataset of erroneous reasoning chains paired with corrected solutions. This taught the model to reflect on and fix its own errors during generation. More broadly, research on reflection and self-correction in LLMs highlights several strategies that could inspire our training-free framework. One line of work leverages *external critics or verifier models* to provide feedback on intermediate reasoning steps, guiding the model away from incorrect trajectories (An et al. (2023); Li et al. (2023); Shinn et al. (2023)). Another line explores *intrinsic self-correction*, where models are fine-tuned on specially constructed datasets that pair erroneous reasoning trajectories with their corrections (Weng et al. (2023); Yang et al. (2025); Zhang et al. (2024); Han et al. (2024); Yan et al. (2024); Tong et al. (2024); Renze (2024)). To collect such data, prior studies often introduce errors by raising the sampling temperature or by sampling across multiple models, ensuring that the training set contains both flawed and corrected reasoning paths (Xi et al. (2024)). These approaches enable models to revise their reasoning, and prevent error accumulation. Although our method does not involve weight updates, we draw inspiration from these techniques. In particular, the idea of coupling flawed reasoning with reflection and correction motivates our *Error Notebook* design. Instead of using fine-tuning to encode these revision patterns into the model parameters, we operationalize them at inference time: by retrieving analogous past samples and their corrections, we provide the model with direct exemplars of reflection, thereby encouraging more reliable reasoning without any additional training cost.

As shown in Figure 1, we introduce a novel inference-phase strategy for vision-language part retrieval in 3D CAD assemblies. Rather than training or fine-tuning a new model, our approach enhances reasoning on-the-fly through retrospective error analysis and retrieval-augmented guidance. Central to our method is the *Error Notebook*, a mechanism that refines model reasoning at inference time by recording and organizing corrected reasoning trajectories. For each new assembly query, we retrieve analogous cases from the *Error Notebook* and provide them as few-shot exemplars to guide the model's chain-of-thought (CoT) using a retrieval-augmented generation (RAG) strategy. The grammar-constraint (GC) verifier leads to further performance gains on the part retrieval task. We evaluate several state-of-the-art VLMs (including GPT-4 variants and Gemini models) and open-source small VLMs on our benchmark. In summary, our contributions are as follows:

(1) We propose a training-free reasoning framework that combines the *Error Notebook* and RAG for VLM inference. Importantly, our method surpasses traditional training-free inference-time approaches (standard few-shot, self-consistency) and further demonstrates strong improvements even on open-source models (e.g., Qwen2-VL-2B-Instruct and Aya-Vision-8B).

(2) We introduce a grammar-constraint (GC) verifier to ensure the structural validity of corrected CoTs used in the *Error Notebook*. This consistently improves the quality of retrieved exemplars, yielding further gains across all evaluated VLMs.

(3) We reconstruct a multimodal CAD assembly dataset with relational specifications and human-preference annotations, consisting of 752 assemblies with part counts ranging from 2 to 249.

(4) From the perspective of the engineering value, we design an effective two-stage VLM strategy that first generates part descriptions and then uses these descriptions for retrieval, thereby overcoming the challenge of processing extremely long STEP file inputs.

## 2 METHODOLOGY

Section 2.1 provides the problem formulation and two-stage (1st and 2nd VLM) process of our part retrieval framework, as illustrated in Figure 1. Section 2.2 describes the dataset construction and human preference annotation process. We introduce the construction of the *Error Notebook* in Section 2.3, which is used for the 2nd VLM reasoning. We further describe the GC mechanism for refining the *Error Notebook* in Section 2.4. Finally, the complete inference procedure that integrates the *Error Notebook* with RAG is presented in Section 2.5 and summarized in Figure 2 (b).

### 2.1 PROBLEM OVERVIEW AND PROPOSED PART RETRIEVAL FRAMEWORK

Given a 3D CAD assembly $\mathcal{A}$ consisting of $n$ parts $\{P_1, P_2, \ldots, P_n\}$, and a natural language assembly specification $S$, our goal is to retrieve the subset of parts $\mathcal{P}^* \subseteq \{P_1, \ldots, P_n\}$ that satisfy the specified relation described in $S$. The retrieval process is formulated as a two-stage VLM reasoning pipeline:

**Stage 1 (1st VLM): Part Description Generation.** For each part $P_i$ in an assembly, we provide both the image of the complete assembly $\mathcal{I}_{\text{assembly}}$ and the image of the individual part $\mathcal{I}_{P_i}$ as input to a model $f_{\text{desc}}(\cdot)$. The decomposition of the assembly into individual components is implemented using Python libraries, as illustrated in Figure A.7. The model is prompted to generate a concise and discriminative sentence $d_i$ describing $P_i$ with respect to the assembly context:

$$d_i = f_{\text{desc}}(\mathcal{I}_{\text{assembly}}, \mathcal{I}_{P_i}, \text{prompt}_{\text{desc}}), \tag{1}$$

where $\text{prompt}_{\text{desc}}$ is a designed instruction that encourages the model to focus on salient geometric and functional features. Repeating this process for all parts yields a set of part-level descriptions $\{d_i\}_{i=1}^{N}$, which are organized into a structured JSON with one entry per part, where the number of entries equals the number of parts in the assembly.

**Stage 2 (2nd VLM): Specification-Aware Part Retrieval via CoT Reasoning.** We first aggregate the part-level outputs from *Stage 1* into a structured mapping (JSON) from part filenames (identifiers) to their corresponding descriptions, denoted as $\mathcal{D} = \{filename_i : d_i\}_{i=1}^{n}$. Given the assembly image $\mathcal{I}_{\text{assembly}}$, the aggregated description mapping $\mathcal{D}$, and the specification $S$, we prompt the model $f_{\text{retr}}(\cdot)$ to identify the relevant parts:

$$\hat{\mathcal{P}}^* = f_{\text{retr}}(\mathcal{I}_{\text{assembly}}, \mathcal{D}, S, \text{prompt}_{\text{retr}}), \tag{2}$$

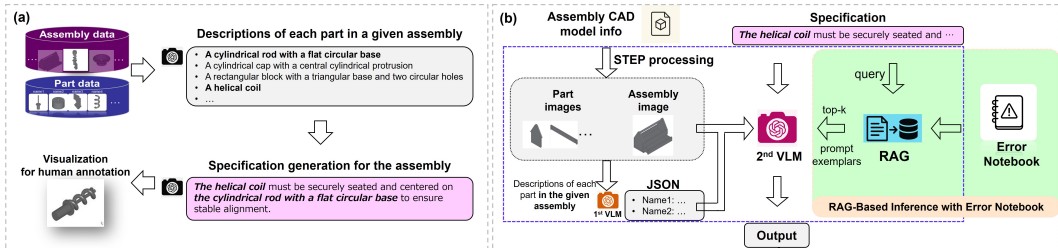

Figure 2: **Overview of the (a) dataset construction pipeline and (b)** *Error Notebook* **+ RAG-based inference process.** (a) For each assembly, a VLM is used to generate concise and discriminative natural language descriptions for every part. Subsequently, the model generates assembly-level specification sentences describing the required relationship. To support human annotation, the specified parts are merged and visualized as a CAD model image. (b) Following the 1st VLM, at the 2nd stage, given the assembly specification, the system retrieves the most relevant examples from the *Error Notebook* according to the assembly specification, incorporates these as few-shot exemplars, and then performs step-by-step reasoning to generate the final answer.

where $\text{prompt}_{\text{retr}}$ requires the model to reason step-by-step (CoT) and produce both an interpretable rationale and the final answer in the form of a subset of part filenames.

## 2.2 DATASET CONSTRUCTION AND HUMAN ANNOTATION

Our study is based on the Fusion 360 Gallery Dataset (Willis et al., 2021b;a; Lambourne et al., 2021)). Specifically, we utilize the *Assembly Dataset*, a subset of the Fusion 360 Gallery Dataset, which comprises multi-part CAD assemblies enriched with detailed information regarding joints, contact surfaces, and holes. For this work, we focus on archive *a1.0.0_00*, which contains 752 assemblies (the Fusion 360 Assembly Dataset is divided into multiple sets whose assembly counts, and CAD model types are highly consistent). Each assembly project within this archive includes a single assembly and the corresponding part information (such as PNG images, STEP files, and additional metadata). The PNG files provide 2D image representations of the 3D models. STEP files (Standard for the Exchange of Product model data), as defined by ISO 10303, are neutral file formats that facilitate the exchange of 3D model data across different CAD software platforms, preserving geometry, structure, and other essential attributes.

Figure 2 (a) shows the overview of the dataset construction pipeline. To begin, we catalog all part names and count the number of parts per assembly. Next, we utilize the GPT-4o (Omni) to generate concise and descriptive noun phrases for each individual part. For each part, we provide both the overall assembly image and the part image as input, so that the model can generate the part description with full awareness of the assembly context. Each phrase is intended to succinctly describe the part's primary shape and distinguishing features, thereby allowing it to be differentiated from other parts within the same assembly. We provide several few-shot examples to guide the model toward generating higher-quality descriptions. Figure A.3 presents the prompt.

Subsequently, we leverage GPT-4o (Omni) again to further generate high-level specifications for the 3D assemblies. Each specification is focused on relationships between selected parts within the assembly. The process is as follows: First, the model reviews the assembly image and the corresponding list of part descriptions. It then selects two part descriptions that are most likely to exhibit a direct physical, spatial, or functional relationship (e.g., fit, mounting, alignment, or coupling). For each pair, the model generates a specification sentence that articulates the relationship, fit, or assembly condition between the two parts. The resulting set of filenames, $f_i$, is subsequently adopted as the ground truth for downstream part retrieval tasks. Figure A.4 presents the prompt for this process.

Finally, to facilitate the construction of a human preference database, we incorporate a human annotation stage. Each annotation bundle includes the merged part image, the original assembly image, and the relevant specification sentence. Professional annotators review and filter items according to the following procedure:

(1) Examine the assembly image to gain a comprehensive understanding of the overall structure.

(2) Items with overly similar part descriptions are discarded, as such cases can lead to ambiguity and multiple possible answers during part retrieval.

(3) Assemblies in which the overall structure is nearly indistinguishable from one or more of its constituent parts are also filtered out.

(4) Any other scenarios that may introduce ambiguity or permit multiple correct answers in part retrieval are excluded.

## 2.3 ERROR NOTEBOOK CONSTRUCTION

To further improve model reasoning, we construct an *Error Notebook* for the 2nd VLM that leverages the ability of VLMs to self-reflect and correct mistakes within their step-by-step reasoning process. Given, for each assembly, the assembly image $\mathcal{I}_{\text{assembly}}$, the mapping from part filenames to their descriptions $\mathcal{D}$, a specification $S$, the previous CoT reasoning $R^{\text{prev}}$, and the ground-truth filenames $\mathcal{P}^{*(\text{gt})}$. The goal is to generate a **corrected reasoning trajectory** $R^{\text{corr}}$ that leads to the correct solution, as shown in Figure 3.

In theory, we formalize the step-by-step reasoning process as a trajectory $R = (s_1, s_2, \ldots, s_n, \hat{a})$, where $s_i$ are intermediate reasoning steps and $\hat{a}$ is the predicted answer. A suboptimal trajectory, $R^{\text{prev}}$, may contain both correct steps and erroneous steps, ultimately leading to an incorrect prediction. Models are expected to identify and revise the first erroneous step in $R^{\text{prev}}$. We define a *corrected reasoning trajectory* $R^{\text{corr}}$ as the concatenation of: 1) all steps up to the first error, 2) a natural language reflection that pinpoints and transitions from the error, and 3) the corrected reasoning steps that ultimately yield the ground-truth answer $\mathcal{P}^{*(\text{gt})}$. Formally, if $R^{\text{prev}} = (s_1^g, \ldots, s_k^g, s_1^b, \ldots, s_m^b, a^b)$, where $s_i^g$ are correct steps and $s_j^b$ are erroneous, we extract the subsequence ending at the first error, $R_{\text{sub}}^{\text{prev}} = (s_1^g, \ldots, s_k^g, s_1^b)$. The corrected trajectory is then constructed as:

$$R^{\text{corr}} = R_{\text{sub}}^{\text{prev}} \oplus \text{TR} \oplus R^g, \tag{3}$$

where TR is a transition phrase, and $R^g$ is the correct trajectory from the correction point to the ground-truth answer $\mathcal{P}^{*(\text{gt})}$.

In our approach,

$$R^{\text{corr}} = f_{\text{corr}}\big(\mathcal{I}_{\text{assembly}}, \mathcal{D}, S, R^{\text{prev}}, \mathcal{P}^{*(\text{gt})}, \text{prompt}_{\text{corr}}\big). \tag{4}$$

The $\text{prompt}_{\text{corr}}$ instructs the model to: (1) Read and follow the previous reasoning $R^{\text{prev}}$ step by step. (2) Upon encountering the first logical or factual error, stop and explicitly articulate the transition. (3) From that point onward, independently correct the error, reasoning step by step until reaching $\mathcal{P}^{*(\text{gt})}$. (4) If no errors are detected, simply reproduce the previous correct reasoning and answer.

## 2.4 VERIFYING CORRECTED REASONING: GC FILTERING

**To ensure that the corrected trajectories included in the *Error Notebook* are logically well-formed, we introduce a *grammar-constraint (GC)* check / filtering mechanism.** This procedure serves as a deterministic verifier that inspects each corrected reasoning trace and determines whether it satisfies a set of structural and semantic validity conditions.

Given a corrected reasoning trajectory $R^{\text{corr}}$ and the set of allowable part filenames $\mathcal{P}$, we check whether the final segment of $R^{\text{corr}}$ contains a well-defined and valid answer. Concretely, the verifier searches for a line beginning with the phrase *Final Answer:* and extracts the predicted filenames. A reasoning trace is accepted if and only if (1) such a line exists, (2) at least one filename is provided, and (3) every predicted filename appears in the allowed set $\mathcal{P}$. In practice, we evaluate two variants of this filtering rule:

**Strict grammar constraint (sGC).** This variant requires the explicit presence of a *Final Answer:* line and accepts a corrected trajectory only if it satisfies all structural validity rules.

**Relaxed grammar constraint (rGC).** To accommodate models whose corrected reasoning is logically sound but omits the explicit *Final Answer:* marker, we introduce a relaxed variant that additionally accepts trajectories that are identical to sGC except for missing this indicator.

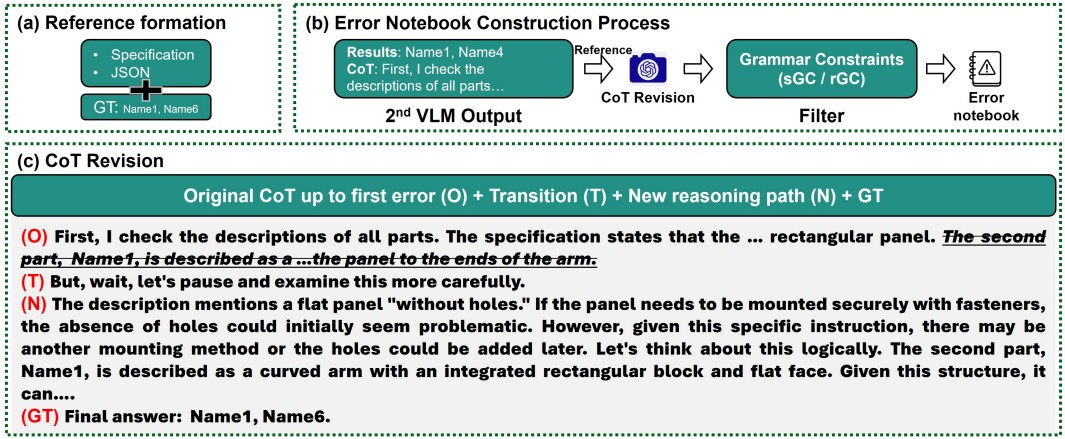

Figure 3: *Error Notebook* **construction.** We define a corrected reasoning trajectory as the concatenation of: 1) all steps up to the first error, 2) a natural language reflection that pinpoints and transitions from the error, and 3) the corrected reasoning steps that ultimately yield the ground-truth answer. The proposed GC check is further employed to improve the quality of the *Error Notebook*.

## 2.5 ERROR NOTEBOOK + RAG-BASED INFERENCE

In the inference stage, we adopt a RAG strategy that leverages examples from the *Error Notebook* as few-shot exemplars. Specifically, it retrieves the top-$n$ most relevant samples from the *Error Notebook* based on their similarity to the current assembly specification, using the corrected CoT trajectories from these entries to guide the model's reasoning. Figure 2 (b) shows the overview of the overall inference process based on VLMs.

Given an instance defined by the assembly image $\mathcal{I}_{\text{assembly}}$, the mapping from part filenames to descriptions $\mathcal{D}$, and a specification $S$, the RAG-based inference proceeds as follows:

(1) **Sample Retrieval:** Let $\mathcal{E} = \{e_1, \ldots, e_M\}$ denote the set of entries in the *Error Notebook*, each comprising a specification $S_j$, part descriptions $\mathcal{D}_j$, and a corrected CoT trajectory $R_j^{\text{corr}}$. For the current query, compute the similarity $\text{sim}(S, S_j)$ between $S$ and each $S_j$ in $\mathcal{E}$. **To avoid data leakage, the current query instance $e_{\text{cur}}$ is excluded from retrieval and will never appear among its own few-shot exemplars.** The top-$n$ most similar samples are selected:

$$\{e_{k_1}, \ldots, e_{k_n}\} = \arg \max_{e_j \in \mathcal{E} \setminus \{e_{\text{cur}}\}} \text{sim}(S, S_j), \tag{5}$$

where $e_{\text{cur}}$ denotes the current query instance.

(2) **Few-Shot Prompt Construction:** For each retrieved sample $e_{k_i}$, construct a prompt block containing the assembly context, part descriptions, specification, and the corrected CoT $R_{k_i}^{\text{corr}}$ with the corresponding final answer. These prompt blocks are concatenated to serve as few-shot exemplars for the current query.

(3) **Main Query Prompt:** The final model input consists of (i) the few-shot exemplars constructed above and (ii) the current query context, which includes the assembly image $\mathcal{I}_{\text{assembly}}$, part descriptions $\mathcal{D}$, and specification $S$. The model is prompted to perform step-by-step reasoning, leveraging the retrieved exemplars as references.

Formally, let $F$ denote the few-shot prompt constructed from the top-$n$ retrieved entries. The model's output is given by:

$$R = f_{\text{rag}}\big(F, \mathcal{I}_{\text{assembly}}, \mathcal{D}, S, \text{prompt}_{\text{main}}\big), \tag{6}$$

where $R$ is the model's answer, and $\text{prompt}_{\text{main}}$ provides the instructions for the inference task.

## 3 EXPERIMENTS

### 3.1 IMPLEMENTATION DETAILS

Our pipeline interacts with VLMs (e.g., GPT-4o, Gemini) via API endpoints. For each inference call, images are encoded as base64 data URLs. We implement error handling with exponential backoff and up to 3 retries in the event of API errors. To process the dataset efficiently, all major computation steps are parallelized for asynchronously executing functions using multiple threads. Each assembly is processed as an independent unit. The generated part descriptions, which serve as intermediate outputs, are stored in JSON format. For fair comparison, both/all experiments on the same model/group (experiments w/o E-Notebook and w/ E-Notebook) employ identical description JSON files. Unless otherwise specified, the value of $k$ for RAG's top-$k$ retrieval is equal to the number of exemplars in the part retrieval stage, which defaults to 2.

### 3.2 MAIN RESULT

Table 1: Accuracy comparison of general models with and without *Error Notebook-RAG* integration on self-generated and human preference datasets. The best result is highlighted in **bold**. We divided the data from both datasets into 4 groups based on the number of parts in each assembly, reflecting the varying difficulty levels.

| Strategy | Self-generated dataset | | | | | Human preference dataset | | | | |
|---|---|---|---|---|---|---|---|---|---|---|
| | Overall | $< 10$ | $10 - 20$ | $20 - 50$ | $> 50$ | Overall | $< 10$ | $10 - 20$ | $20 - 50$ | $> 50$ |
| **GPT-4o (Omni)** | | | | | | | | | | |
| w/o E-Notebook | 28.5 | 40.7 | 22.4 | 15.3 | 5.0 | 41.7 | 47.9 | 32.4 | 26.5 | 0.0 |
| w/ E-Notebook | 48.3 | 66.8 | 35.9 | 29.7 | **16.3** | 65.1 | 75.5 | 42.6 | 41.2 | 21.4 |
| w/ E-Notebook+sGC | **48.5** | **67.0** | **36.5** | **32.2** | 12.5 | **66.8** | **75.5** | **48.5** | **50.0** | **21.4** |
| **GPT-4o mini** | | | | | | | | | | |
| w/o E-Notebook | 13.6 | 20.5 | 10.9 | 4.2 | 1.3 | 19.3 | 24.8 | 10.3 | 0.0 | 0.0 |
| w/ E-Notebook | 24.9 | 34.9 | **25.0** | 5.9 | **7.5** | 35.4 | 41.5 | 29.4 | 8.8 | 7.1 |
| w/ E-Notebook+sGC | **25.9** | **37.7** | 20.5 | **11.0** | 5.0 | **36.4** | **42.6** | 29.4 | **11.8** | **7.1** |
| **Gemini 2.5 Pro Non-streaming** | | | | | | | | | | |
| w/o E-Notebook | 36.5 | 55.1 | 25.6 | 14.4 | 6.2 | 54.0 | 65.2 | 35.3 | 20.6 | 0.0 |
| w/ E-Notebook | 42.2 | 60.9 | **30.8** | **21.2** | **11.3** | 59.5 | 69.5 | **42.6** | 29.4 | **14.3** |
| w/ E-Notebook+sGC | **42.9** | **64.8** | 28.8 | 20.3 | 5.0 | **62.1** | **74.1** | 38.2 | **32.4** | 7.1 |
| **Gemini 2.0 Flash Non-streaming** | | | | | | | | | | |
| w/o E-Notebook | 30.9 | 46.8 | 21.2 | 12.7 | 5.0 | 44.2 | 53.5 | 23.5 | 20.6 | 14.3 |
| w/ E-Notebook | **40.4** | **58.2** | **31.4** | 19.5 | 8.7 | 56.8 | **67.0** | 39.7 | 23.5 | 14.3 |
| w/ E-Notebook+sGC | 40.3 | 57.3 | 30.1 | **19.5** | **13.8** | **57.0** | 66.3 | **39.7** | **29.4** | **21.4** |
| **Gemini 1.5 Pro Non-streaming** | | | | | | | | | | |
| w/o E-Notebook | 29.9 | 44.3 | 21.2 | 13.6 | 6.2 | 43.0 | 52.1 | 23.5 | 17.6 | 14.3 |
| w/ E-Notebook | 32.4 | 49.3 | 22.4 | 11.9 | 6.2 | 46.7 | 57.1 | 25.0 | **17.6** | 14.3 |
| w/ E-Notebook+sGC | **36.2** | **51.8** | **26.3** | **17.8** | **12.5** | **50.3** | **60.6** | **30.9** | 14.7 | **21.4** |
| **Cloud Vision (Image) + Gemini 2.0 Flash Non-streaming** | | | | | | | | | | |
| w/o E-Notebook | 35.0 | 51.8 | 25.0 | 14.4 | 8.7 | 50.0 | 58.9 | 38.2 | 17.6 | 7.1 |
| w/ E-Notebook | 40.4 | 59.3 | 30.1 | 16.1 | 11.3 | 57.8 | 66.3 | 47.1 | **29.4** | 7.1 |
| w/ E-Notebook+sGC | **43.2** | **63.2** | **32.7** | **17.8** | 11.3 | **62.3** | **73.0** | **48.5** | 20.6 | **14.3** |

**(1) Our experimental results demonstrate that the proposed *Error Notebooks* with RAG framework enhances retrieval accuracy across all evaluated models and assembly complexities, as summarized in Table 1. The performance gains are particularly pronounced on the human preference dataset.** For example, GPT-4o (Omni) improves from 41.7% to 65.1% overall on the human preference dataset, marking an absolute gain of 23.4%, while its performance on the self-generated dataset also rises from 28.5% to 48.3% (+19.8%). Similar trends are observed for other models: GPT-4o mini increases from 19.3% to 35.4% (+16.1%), Gemini 2.0 Flash Non-streaming

Table 2: Ablation study on the number of exemplars retrieved from the *Error Notebook*. We also analyze the effect of excluding explicit CoT reasoning in each exemplar. *CoT Group* indicates that each retrieved exemplar includes explicit step-by-step reasoning, while *Non-CoT Group* omits such reasoning in the exemplars and includes ground truth only. The data from both datasets are divided into four groups based on the number of parts in each assembly, reflecting varying difficulty levels.

| Number of Exemplars | Self-generated dataset | | | | | Human preference dataset | | | | |
|---|---|---|---|---|---|---|---|---|---|---|
| | Overall | $< 10$ | $10-20$ | $20-50$ | $> 50$ | Overall | $< 10$ | $10-20$ | $20-50$ | $> 50$ |
| **Non-CoT Group** | | | | | | | | | | |
| 1 | 49.4 | 69.5 | 37.8 | 27.1 | 13.8 | 69.3 | 80.5 | 50.0 | 38.2 | 14.3 |
| 5 | 50.1 | 70.4 | 38.5 | 29.7 | 11.3 | 69.1 | 79.8 | 51.5 | 41.2 | 7.1 |
| 10 | 50.6 | 69.8 | 37.8 | 32.2 | 16.3 | 70.4 | 79.4 | 55.9 | 44.1 | 21.4 |
| 20 | 50.8 | 69.3 | 42.3 | 32.2 | 11.3 | 69.1 | 77.7 | 60.3 | 38.2 | 14.3 |
| 50 | 52.7 | 72.0 | 42.3 | 32.2 | 16.3 | 72.9 | 83.0 | 57.4 | 41.2 | 21.4 |
| **CoT Group** | | | | | | | | | | |
| 1 | 49.7 | 68.4 | 39.7 | 30.5 | 12.5 | 67.8 | 77.7 | 54.4 | 38.2 | 7.1 |
| 5 | 49.4 | 67.0 | 38.5 | 32.2 | 16.3 | 67.8 | 75.5 | 52.9 | 50.0 | 28.6 |
| 10 | 49.4 | 66.5 | 42.3 | 29.7 | 15.0 | 68.8 | 76.2 | 61.8 | 44.1 | 14.3 |
| 20 | 51.7 | 69.0 | 42.3 | 35.6 | 16.3 | 71.1 | 79.8 | 57.4 | 52.9 | 7.1 |
| 50 | 49.5 | 67.9 | 37.8 | 33.1 | 13.8 | 68.1 | 77.0 | 51.5 | 52.9 | 7.1 |

from 44.2% to 56.8% (+12.6%), and Gemini 1.5 Pro Non-streaming from 43.0% to 46.7% (+3.7%). Another clear trend is that improvements are not limited to small assemblies: while the largest absolute gains often appear in cases with fewer parts (e.g., $< 10$ parts, GPT-4o Omni rises from 47.9% to 75.5%), consistent accuracy improvements are observed across all part-count intervals, including the more challenging $> 50$ parts group. These results highlight the effectiveness and generality of the proposed *Error Notebooks* + RAG strategy, which enhances inference across different proprietary (GPT, Gemini) models, without requiring additional training.

**The effect of GC check on Table 1.** We then rebuilt the *Error Notebook* using entries that passed this strict grammar constraints (sGC) check, and re-ran inference with the same RAG pipeline. And this trick further produces up to 4.5 points of improvement on the human preference dataset.

While Table 1 demonstrates the performance gap between models with and without *Error Notebooks*, Table 2 further shows that once *Error Notebooks* are incorporated, the **number** of exemplars retrieved by RAG has only a minor effect on final accuracy. For instance, on the self-generated dataset, the overall accuracy of the Non-CoT group varies only slightly between 49.4% (1 exemplar) and 52.7% (50 exemplars). A similar trend holds for the CoT group, where performance remains stable in the narrow range of 49.4% to 51.7%. Consistent patterns are observed on the human preference dataset. These results indicate that the key factor driving improvements is the presence of *Error Notebooks* themselves, and the effect of the specific number of exemplars sampled is negligible.

**(2) The results in Table 2 and Figure A.1 show that incorporating CoT reasoning from the *Error Notebook* is particularly valuable for challenging cases with higher part counts ($> 10$).** For assemblies with fewer parts ($< 10$), the Non-CoT group, where only final answers are given, often performs comparably or even slightly better, suggesting that in simple scenarios, direct access to the final correct solution is sufficient. By contrast, for complex assemblies with 10–50 parts, the CoT group consistently outperforms the Non-CoT group across nearly all exemplar sizes, confirming that step-by-step reasoning provides crucial guidance for harder queries. This trend is observed across all exemplar group sizes, with one notable exception: when using 50 exemplars, the CoT group shows a drop in accuracy. We attribute this to excessively long prompts caused by concatenating many CoTs, which may interfere with the model's judgment. A second important observation (Figure A.1) is that for simple assemblies, increasing the number of exemplars has little effect, regardless of whether CoT is used. In contrast, for complex assemblies, accuracy improves as the number of exemplars increases, up to around 20 exemplars.

**(3) The ablation experiments on Table 3 show that our method outperforms two traditional training-free, inference-time approaches.** We conducted ablation experiments to compare our *Error Notebook* method with two representative training-free, inference-time approaches. The ex-

Table 3: Ablation comparison between training-free baselines and our proposed method.

| Strategy | Self-generated dataset | | | | | Human preference dataset | | | | |
|---|---|---|---|---|---|---|---|---|---|---|
| | Overall | < 10 | 10–20 | 20–50 | > 50 | Overall | < 10 | 10–20 | 20–50 | > 50 |
| Standard few-shot | 26.6 | 37.4 | 19.2 | 16.9 | 6.2 | 37.7 | 42.9 | 29.4 | 17.6 | 21.4 |
| w/o E-Notebook | 28.5 | 40.7 | 22.4 | 15.3 | 5.0 | 41.7 | 47.9 | 32.4 | 26.5 | 0.0 |
| Self-consistency | 38.9 | 54.6 | 30.1 | 21.2 | 11.3 | 54.8 | 61.7 | 42.6 | 29.4 | 35.7 |
| w/ E-Notebook (ours) | **48.3** | **66.8** | **35.9** | **29.7** | **16.3** | **65.1** | **75.5** | **42.6** | **41.2** | **21.4** |

perimental settings are as follows. For **standard few-shot learning**, we use GPT-4o (Omni) with 2 API endpoints, and adopt two GPT-generated exemplars as few-shot examples (aligned with the 2-exemplar setting in Table 1). We keep the full two-stage pipeline: the 1st VLM generates part descriptions from the assembly and part images; the second VLM performs reasoning. Standard few-shot is applied to the 2nd VLM (reasoning stage). For **self-consistency**, we keep the same two-stage VLM pipeline. The 1st VLM generates part-level descriptions exactly as in our main method. For the 2nd VLM, we replace the *Error Notebook* with a self-consistency strategy: GPT-4o (Omni), temperature 0.7, 5 independent samples, followed by majority voting. **Across both datasets, our method consistently outperforms those baselines**.

Table 4: Results of Qwen2-VL-2B-Instruct. We report both accuracy and the number of correctly solved cases (in parentheses) under identical settings as Table 1.

| Strategy | Self-generated dataset | | | Human preference dataset | |
|---|---|---|---|---|---|
| | Overall | < 10 (361) | 10–20 (156) | Overall | < 10 (282) |
| w/o E-Notebook | 0.8 (6) | 1.7 (6) | 0.0 (0) | 1.5 (6) | 2.1 (6) |
| w/ E-Notebook | 6.4 (46) | 12.5 (45) | 0.6 (1) | 10.8 (43) | 15.2 (43) |
| **Improvement** | **+5.6 (+40)** | **+10.8 (+39)** | **+0.6 (+1)** | **+9.3 (+37)** | **+13.1 (+37)** |
| w/ E-Notebook+sGC | 3.6 (26) | 7.2 (26) | 0.0 (0) | 6.0 (24) | 8.5 (24) |
| w/ E-Notebook+rGC | 6.6 (47) | 12.7 (46) | 0.6 (1) | 10.8 (43) | 15.2 (43) |
| w/ gE-Notebook+sGC* | 8.4 (60) | 16.6 (60) | 0.0 (0) | 14.6 (58) | 20.6 (58) |
| **Improvement (* - w/o)** | **+7.6 (+54)** | **+14.9 (+54)** | **+0.0 (+0)** | **+13.1 (+52)** | **+18.5 (+52)** |

**(4) Our method also demonstrates strong performance on *open-source* models.** We further evaluated our approach on two open-source VLMs, **Qwen2-VL-2B-Instruct** (Wang et al., 2024) and **Aya-Vision-8B** (Dash et al., 2025). All experimental settings (prompting format, RAG retrieval, and evaluation protocol) were kept identical to those used in Table 1. For Qwen2-VL-2B-Instruct, experiments were conducted on $8 \times$ A40 GPUs for approximately 3 days. A detailed breakdown of its performance is reported in Table 4, and the results for Aya-Vision-8B appear in Appenidx A.2.

During the *grammar-check filtering* evaluation, we compared three variants. The *E-Notebook+sGC* configuration applies the same strict rule used for proprietary models. However, we found that the 2B model frequently produced otherwise valid reasoning traces that lacked the explicit *Final Answer:* marker, causing many acceptable traces to be discarded. This substantially reduced the size of the *Error Notebook* and degraded performance. The *E-Notebook+rGC* variant therefore relaxes this requirement, leading to improved accuracy compared to the basic *E-Notebook* setup. Finally, the *gE-Notebook+sGC* variant uses an *Error Notebook constructed entirely from GPT-4o (Omni)* while still performing inference with the 2B model, reinstating the strict grammar rule under this cross-model setting. **Strikingly, the cross-model variant (*gE-Notebook+sGC*) achieves the strongest performance across all configurations.** On the human-preference dataset, the 2B model equipped with *gE-Notebook+sGC* performs only **4.2 points below GPT-4o mini** in the <10 group. These results indicate that a lightweight open-source model, when paired with a high-quality *Error Notebook* and appropriate grammar-check strategies, can closely approach the performance of substantially stronger proprietary VLMs.

Overall, these findings confirm that **the *Error Notebook* framework provides substantial and meaningful gains for open-source VLMs**. Moreover, the improvements achieved through cross-model distillation show that the *Error Notebook* can serve as an effective mechanism for transferring high-quality reasoning traces from powerful proprietary models to compact open-source ones **without any finetuning or additional training**.

## 3.3 EFFICIENCY ANALYSIS

**Token Usage and Latency.** We conducted a runtime and token-cost evaluation on 100 samples under: GPT-4o (Omni), a single API endpoint, one worker, and no batching. Table 5 summarizes the results. Although using the *Error Notebook* increases prompt tokens, **inference does not become slower** (8.04s vs 6.50s). Corrected exemplars may improve reasoning coherence and reduce internal search depth. The one-time correction step is lightweight (7.39 s per sample). Also, 1st VLM latency is high since it depends on the **number** of the CAD model's parts. Overall, the *Error Notebook* introduces no prohibitive overhead, and RAG-enhanced inference remains efficient.

Table 5: Latency and token usage for constructing the *Error Notebook* and performing inference.

| Setting | Avg Time (s) | Prompt Tokens | Completion Tokens |
|---|---|---|---|
| 1st VLM (part description) | 78.32 | – | – |
| **2nd VLM (w/o E-Notebook)** | 8.04 | 967.7 | 235.4 |
| **2nd VLM (w/ E-Notebook)** | 6.50 | 1815.3 | 278.7 |
| CoT Correction Step | 7.39 | 1328.7 | 377.5 |

**API Call Cost.** The total number of VLM calls required to construct the *Error Notebook* over $n$ samples is:

$$\sum_{i=1}^{n}(\text{part count}_i + 1) + n \times 1. \tag{7}$$

For each sample $i$, part count$_i$ VLM calls are used to generate part-level descriptions, plus **one** call for the (initial) part retrieval result. Then, new CoTs must be generated for correction, adding one more call per sample.

## 4 CONCLUSION

In this work, we introduced a novel *Error Notebook*-guided, training-free part retrieval approach for complex 3D CAD assemblies. Our framework leverages retrospective error analysis and RAG to enhance VLM reasoning without additional training or fine-tuning. By systematically constructing *Error Notebooks* that capture and correct flawed reasoning trajectories, and by retrieving specification-similar exemplars at inference time, our method consistently improves accuracy across multiple proprietary VLMs. Importantly, our method surpasses traditional training-free inference-time approaches (standard few-shot, self-consistency) and further demonstrates strong improvements even on open-source models (e.g., Qwen2-VL-2B-Instruct and Aya-Vision-8B).

Future work will explore cross-domain applications of *Error Notebooks*, aiming to establish a more general paradigm for training-free reflective reasoning in multimodal AI.

## 5 THE USE OF LARGE LANGUAGE MODELS (LLMS)

We used LLMs as the experiment subject to study the improvement of our method on existing LLMs. We also used LLMs to polish writing.

## 6 ETHICS STATEMENT

Our dataset construction process relies on professional human annotators, who were compensated fairly and provided clear annotation guidelines. Care was taken to exclude ambiguous or misleading cases to avoid introducing bias into the dataset. No personally identifiable information or sensitive

data is involved. The proposed methods are intended for engineering and design applications, such as automated verification in CAD workflows, and do not pose foreseeable risks of misuse.

## 7 REPRODUCIBILITY STATEMENT

All code for dataset preprocessing, part description generation, *Error Notebook* construction, and inference experiments is provided in the Supplementary Material. Detailed descriptions of dataset construction (including filtering and annotation protocols) are provided in Section 2.2. Experimental settings, including API interaction details, hyperparameters, and error-handling mechanisms, are documented in Section 3.1. Reproduction of our results requires access to the Fusion 360 Gallery Assembly dataset and VLM APIs (e.g., GPT-4o, Gemini).

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

# A APPENDIX

## A.1 ABBREVIATIONS

| | |
|---|---|
| VLM | Vision-Language Model |
| LLM | Large Language Model |
| CAD | Computer-Aided Design |
| STEP | STandard for the Exchange of Product model data (ISO 10303) |
| CoT | Chain-of-Thought |
| RAG | Retrieval-Augmented Generation |
| API | Application Programming Interface |
| GT | Ground Truth |
| GPT | Generative Pre-trained Transformer |
| **GC** | Grammar Constraint |
| sGC | strict Grammar Constraint |
| rGC | relaxed Grammar Constraint |
| **E-Notebook** | Error Notebook |
| w/o | without |
| w/ | with |

## A.2 SUPPLEMENTARY RESULTS

**(1) Retrieval relevance to Table 1.** We report the *retrieval relevance* results as shown in Table A.1. Define TP = |GT ∩ Pred|, FP = |Pred − GT|, FN = |GT − Pred|, then:

$$\text{Recall(R)} = \frac{\text{TP}}{\text{TP} + \text{FN}}, \tag{A.1}$$

$$\text{Precision(P)} = \frac{\text{TP}}{\text{TP} + \text{FP}}, \tag{A.2}$$

$$\text{F1} = \frac{2PR}{P + R}. \tag{A.3}$$

We report both **global averaged Recall/F1** and **per-group Recall/F1** based on the number of parts ($< 10$, $10$–$20$, $20$–$50$, $> 50$), evaluated on the self-generated dataset. **In Table A.1, we can see that the proposed *Error Notebook* method consistently yields clear and meaningful improvements in retrieval relevance.**

**(2) Our method is not highly sensitive to the specific retrieval scoring function.** In Table 1, the *Error Notebook* relies on a *character-level similarity retriever*, which computes a normalized character-level matching score between textual specifications. To further examine whether our method is sensitive to the retrieval scoring function, we additionally implemented a new retriever based on *token-level Jaccard similarity* as shown in Table A.2. This new version tokenizes each specification and measures the overlap between the resulting token sets. Overall, the token-level Jaccard retriever yields slightly higher accuracy (approximately +2% on the self-generated dataset). Importantly, for both retriever methods, the ***Error Notebook* consistently provides large and robust gains** over the baseline.

Table A.1: Retrieval relevance evaluation.

| Strategy | Global Recall | Global F1 | Per-group Recall / F1 | | | |
|---|---|---|---|---|---|---|
| | | | < 10 | 10–20 | 20–50 | > 50 |
| **GPT-4o (Omni)** | | | | | | |
| w/o E-Notebook | 0.406 | 0.532 | 0.520 / 0.664 | 0.362 / 0.481 | 0.277 / 0.370 | 0.171 / 0.239 |
| w/ E-Notebook | **0.692** | **0.686** | 0.828 / 0.837 | 0.644 / 0.629 | 0.557 / 0.534 | 0.367 / 0.364 |
| **GPT-4o mini** | | | | | | |
| w/o E-Notebook | 0.261 | 0.385 | 0.344 / 0.494 | 0.218 / 0.325 | 0.179 / 0.269 | 0.089 / 0.144 |
| w/ E-Notebook | **0.500** | **0.523** | 0.619 / 0.675 | 0.500 / 0.504 | 0.289 / 0.288 | 0.272 / 0.275 |
| **Gemini 2.5 Pro Non-streaming** | | | | | | |
| w/o E-Notebook | 0.627 | 0.607 | 0.778 / 0.781 | 0.571 / 0.532 | 0.451 / 0.416 | 0.316 / 0.304 |
| w/ E-Notebook | **0.662** | 0.595 | 0.815 / 0.796 | 0.590 / 0.569 | 0.472 / 0.444 | 0.392 / 0.225 |
| **Gemini 2.0 Flash Non-streaming** | | | | | | |
| w/o E-Notebook | 0.552 | 0.573 | 0.681 / 0.728 | 0.529 / 0.531 | 0.400 / 0.392 | 0.241 / 0.254 |
| w/ E-Notebook | **0.630** | **0.628** | 0.777 / 0.784 | 0.583 / 0.584 | 0.468 / 0.446 | 0.297 / 0.296 |
| **Gemini 1.5 Pro Non-streaming** | | | | | | |
| w/o E-Notebook | 0.565 | 0.554 | 0.717 / 0.727 | 0.522 / 0.497 | 0.366 / 0.340 | 0.253 / 0.247 |
| w/ E-Notebook | **0.575** | **0.557** | 0.745 / 0.738 | 0.474 / 0.456 | 0.396 / 0.362 | 0.272 / 0.261 |
| **Cloud Vision (Image) + Gemini 2.0 Flash Non-streaming** | | | | | | |
| w/o E-Notebook | 0.617 | 0.604 | 0.750 / 0.776 | 0.583 / 0.553 | 0.438 / 0.398 | 0.342 / 0.318 |
| w/ E-Notebook | **0.636** | **0.622** | 0.788 / 0.794 | 0.577 / 0.562 | 0.447 / 0.412 | 0.342 / 0.326 |

Table A.2: Comparison between character-level and token-level retrieval scoring functions.

| Strategy | Self-generated dataset | | | | | Human preference dataset | | | | |
|---|---|---|---|---|---|---|---|---|---|---|
| | Overall | < 10 | 10–20 | 20–50 | > 50 | Overall | < 10 | 10–20 | 20–50 | > 50 |
| w/o E-Notebook (Table 1) | 28.5 | 40.7 | 22.4 | 15.3 | 5.0 | 41.7 | 47.9 | 32.4 | 26.5 | 0.0 |
| **w/ E-Notebook (Table 1, character-level)** | **48.3** | 66.8 | 35.9 | 29.7 | 16.3 | **65.1** | 75.5 | 42.6 | 41.2 | 21.4 |
| **w/ E-Notebook (New, token-level)** | **50.2** | 68.4 | 39.7 | 31.4 | 16.3 | **68.1** | 77.3 | 50.0 | 47.1 | 21.4 |

**(3) The results in Figure A.1 show that incorporating CoT reasoning from the *Error Notebook* is particularly valuable for challenging cases with higher part counts** $(10 - 50)$.

**(4) We demonstrate the effectiveness of the proposed two-stage pipeline.** As shown in Figure A.2, the proposed two-stage pipeline for part retrieval in 3D CAD assemblies achieves significantly higher accuracy compared to the image-only reasoning baseline. In the image-only setup, both the assembly image and individual part images are directly fed to the VLM in a single inference step, relying solely on visual input. In contrast, our proposed method first utilizes the VLM to generate concise part descriptions within the assembly context, and then performs part retrieval as a second reasoning step with the assistance of these textual descriptions. This design introduces an additional layer of interpretability and context-awareness, leading to consistent performance improvements across all part count groups. We slightly modified the prompt content to tailor it for this ablation study, using GPT-4o (Omni) as the model. Quantitatively, the image-only baseline yields an overall accuracy of $15.0\%$ ($107/715$). The proposed pipeline achieves an overall accuracy of $33.6\%$ ($240/715$), with $51.2\%$ ($185/361$) for $< 10$ parts, $23.7\%$ ($37/156$) for $10$–$20$ parts, $11.9\%$ ($14/118$) for $20$–$50$ parts, and $5.0\%$ ($4/80$) for $> 50$ parts. These results demonstrate the effectiveness of incorporating part descriptions as intermediate representations.

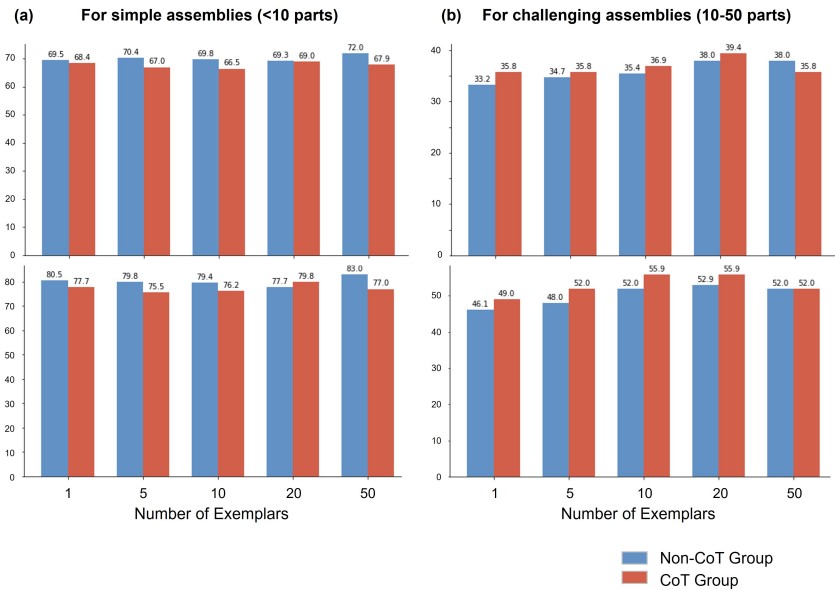

Figure A.1: **Effect of CoT reasoning and exemplar number on retrieval accuracy across different assembly complexities and datasets.** Top row: results on the **self-generated dataset**; bottom row: results on the **human preference dataset**. (a) For simple assemblies ($<$ 10 parts). (b) For more complex assemblies (10–50 parts). The $x$-axis indicates the **number of exemplars retrieved from the *Error Notebook***, where each exemplar consists of either (i) the final corrected answer only (Non-CoT group) or (ii) the corrected CoT reasoning steps plus the final answer (CoT group).

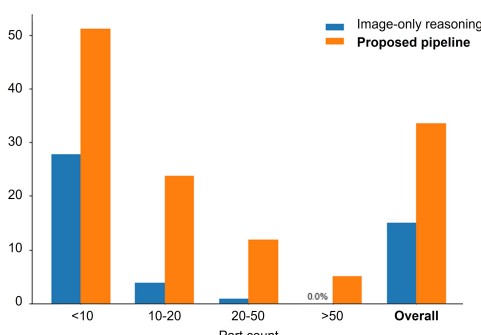

Figure A.2: **Accuracy comparison between proposed pipeline and image-only reasoning.** Performance is shown for the proposed pipeline, which leverages part descriptions as intermediate references, versus the one that directly reasons over images.

**(5) Our method can demonstrate strong performance on *open-source* models.** The results of Aya-Vision-8B is shown in Table A.3. For efficiency, we used 7× A40 GPUs for around 36 hours, and an additional run on 3× H20 GPUs for around 12 hours. All experimental settings (except the model itself) remained identical to those in Table 1. **Therefore, for open-source VLMs, our *Error Notebook* method still brings substantial and meaningful gains.**

Table A.3: Results of Aya-Vision-8B.

| Strategy | Self-generated dataset (666 cases) | | | Human preference dataset (370 cases) | | |
|---|---|---|---|---|---|---|
| | Overall | < 10 | 10–20 | Overall | < 10 | 10–20 |
| w/o E-Notebook | 16 | 16 | 0 | 14 | 14 | 0 |
| w/ E-Notebook (ours) | 54 | 53 | 1 | 51 | 50 | 1 |
| **Improvement** | **+38 (3.4×)** | **+37** | **+1** | **+37 (3.6×)** | **+36** | **+1** |

## A.3 PROMPTS

> You are an expert mechanical engineer. Given Image 1 (the assembly) and Image 2 (an individual part from the assembly), please generate a concise and descriptive noun phrase (not a full sentence). The phrase should briefly describe the part's main shape and any key features, in a way that clearly distinguishes it from the other parts in the assembly. Avoid generic names like 'part' or 'component'. Be specific about the shape and any holes, slots, or functional features. Your output should be a single noun phrase.
>
> ...................................................................................................
>
> For example:
> - A conical mount with a forked top;
> - A cylindrical pin;
> - Two plates with each having holes;
> - A flat round disk with three small holes;
> - A rectangular bracket with two mounting slots.

Figure A.3: Prompt used to generate part-level descriptions in the dataset construction pipeline.

> You are an expert mechanical engineer. Given an image of an assembled product (assembly) and a list of its part descriptions below:
> Part descriptions:
> {desc_list_str}
>
> ...................................................................................................
>
> Your task:
> 1. Review the assembly image and the list of part descriptions.
> 2. Choose any two part descriptions that are most likely to have a direct physical, spatial, or functional relationship in the assembly (such as fit, mounting, alignment, or coupling).
> 3. Generate one specification sentence (inspection/check item) that describes the required relationship, fit, or assembly condition between these two parts, as would appear in a manufacturing or assembly checklist.
> 4. Your specification should be clear, specific, and professional, mentioning both selected part descriptions explicitly.
> 5. Output only one specification sentence. Do not explain your reasoning.
> 6. Output format: The selected two part descriptions (exactly as shown above, separated by a semicolon), then a line break, then the specification sentence.
>
> ...................................................................................................
>
> For example, given descriptions like:
> 1. A cylindrical pin
> 2. A flat plate with holes
> Output:
> A cylindrical pin;A flat plate with holes
> The cylindrical pin must be fully inserted into one of the holes on the flat plate.

Figure A.4: Prompt used to generate specification for each assembly in the dataset construction pipeline.

You are an expert mechanical engineer with a sharp analytical mind. You are given the assembly image, the descriptions of all parts (each as 'filename: description'), the inspection specification, and a previous reasoning process (including its step-by-step thoughts and its Final Answer).

...........................................................................................

Your job:
1. Carefully read the previous reasoning step-by-step. Follow along and reproduce the steps until you encounter the first error or mistake.
2. Once you spot the first mistake, stop following the previous reasoning and use a natural transition phrase (such as: "But, wait, let's pause and examine this more carefully." or "Wait, something seems off. Let's pause and consider what we know so far.") to point out the error and correct it.
3. From that point on, continue the reasoning process in your own words, step-by-step, until you reach the correct answer (i.e., the filenames consistent with the correct ground-truth solution).
4. Do not mention "previous attempt" or "ground-truth solution" explicitly. Make your reasoning sound like a student discovering and correcting their own mistake in real time.
5. If the previous reasoning is already correct, simply reproduce the previous reasoning and the final answer as is.
6. End your output with a "Final Answer:" line followed by the filenames (from the keys above), separated by semicolons (;), with no extra words or punctuation.

Figure A.5: Prompt used to revise CoTs.

Now, for the following question, use the above reasoning as reference and answer step-by-step:
Assembly image:
[image attached]
Part descriptions:{desc_lines}
Specification:{spec}

...........................................................................................

Your task:
1. Think step by step (Chain-of-Thought) and explain how you identify the required part(s).
2. In the last line, write 'Final Answer:' followed by only the selected part filenames (from the keys above), separated by semicolons (;), with no extra words or punctuation.
Example output:
Chain-of-Thought:
First, I check the descriptions of all parts. Only part1.png and part2.png are described as cylindrical pins. Therefore, the required parts are part1.png and part2.png.
Final Answer:
part1.png;part2.png

Figure A.6: Prompt used to generate the part retrieval results.

## A.4 FULL ENGINEERING PIPELINE ILLUSTRATION

To clarify the broader engineering context of our method and help better understand the meaning of *part retrieval* in practical CAD assembly analysis, we provide in Figure A.7 a complete overview of our proposed pipeline in an engineering setting. Specifically, the left side depicts the *STEP processing* stage: an input assembly (in STEP format) is decomposed into its constituent parts using freecad, and subsequently rendered into 2D images using the pythonocc library. This generates intermediate representations (part-level STEP files and rendered images) that provide concrete references for the VLM-based retrieval process. On the right side, a textual specification is provided, and the VLMs enhanced with *Error Notebook* + RAG reasoning produce candidate part identifiers. These are then fused back into the assembly using freecad, and the resulting structure can be visualized with pythonocc.

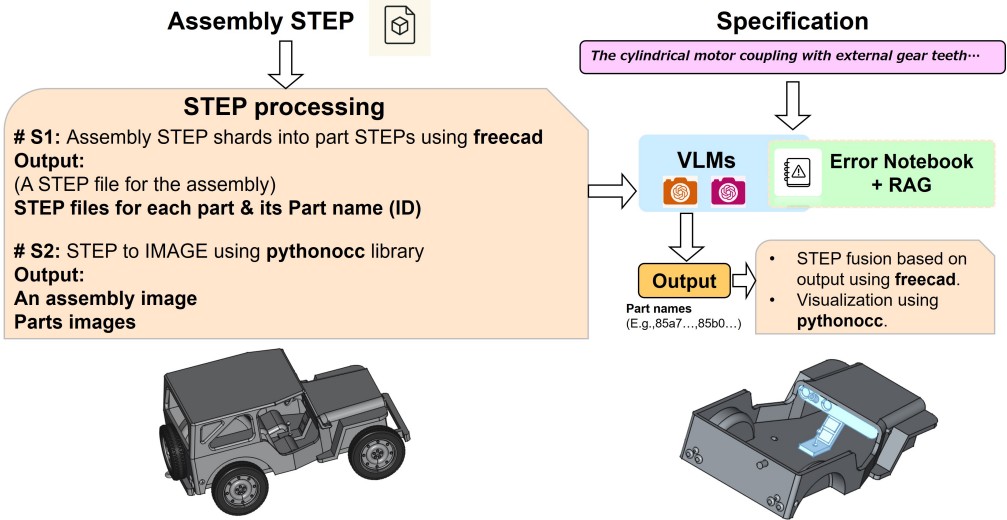

Figure A.7: **Full engineering pipeline for specification-driven part retrieval.** The assembly STEP file is first decomposed into part-level STEP files using freecad, and both the assembly and part images are generated via pythonocc. Given a textual specification, VLMs enhanced with *Error Notebook + RAG* output candidate part identifiers, which are then fused back into the assembly with freecad for visualization.

## A.5 CASE STUDIES

Table A.4: Case studies of assembly-level part retrieval by GPT 4o (Omni) with *Error Notebook*. Each row shows the assembly image, the part count, the specification, and the retrieved results in image format.

| ID | Assembly Image | Part Count | Specification | Retrieval Results |
|---|---|---|---|---|
| 1 |  | 16 | The cylindrical protrusion on the vertical plate must align and securely fit into the curved channel of the rectangular housing. |  |
| 2 |  | 10 | The concave plate with a central circular hole on a short cylindrical base must be securely seated on the cylindrical base with radial grooves, ensuring proper alignment and fit. |  |
| 3 |  | 5 | The curved tapered arm with detailed thumb and fingers must fit snugly within the arm-shaped cavity of the curved block, ensuring full contact and proper alignment. |  |
| 4 |  | 10 | The semi-cylindrical block must fit securely onto the circular grid's central hub without obstructing the radial struts. |  |
| 5 |  | 8 | The cylindrical cap with integrated spout and loop handle must be securely screwed onto the threaded top collar of the cylindrical bottle body, ensuring a leak-proof seal. |  |
| 6 |  | 8 | The curved cylindrical shackle must be securely fitted into one of the lateral round holes on the cylindrical body. |  |

| ID | Assembly Image | Part Count | Specification | Retrieval Results |
|---|---|---|---|---|
| 7 |  | 10 | A flat rectangular plate with diagonal cutouts and rounded corners;A rectangular plate with a larger cutout featuring a stylized raspberry design |  |
| 8 |  | 6 | The helical coil must be securely seated and centered on the cylindrical rod with a flat circular base to ensure stable alignment. |  |
| 9 |  | 5 | The threaded shaft of the knurled cylindrical knob must be securely fastened into the threaded hole of the curved lever arm to ensure proper functionality and alignment of the assembly. |  |
| 10 |  | 9 | The long, curved cylindrical tube must be snugly inserted into the perforated cylindrical opening of the elbow-shaped casing for a secure fit without gaps. |  |
| 11 |  | 4 | The cylindrical rod with a flat end must be fully inserted into the internal square socket of the cylindrical housing, ensuring secure attachment. |  |
| 12 |  | 4 | The hollow cylindrical cap must be securely fitted over the central circular protrusion of the curved base block, ensuring no gaps between the mating surfaces. |  |

