# OpenReview forum: "Error Notebook-Guided, Training-Free Part Retrieval in 3D CAD Assemblies via Vision-Language Models"
_ICLR.cc/2026/Conference — ICLR 2026 Poster_

### Official Review · Reviewer_Hfy1 · 2025-10-30

**Soundness:** 3
**Presentation:** 3
**Contribution:** 3
**Rating:** 6
**Confidence:** 4

**Summary:**

This manuscript discusses the specification aware part retrieval in complex 3D CAD assembly, which is a task that leads to VLM failure due to long context and poor reasoning ability. The author proposes a novel, non training framework that utilizes Error Notebook and RAG to enhance proprietary VLM. This notebook is constructed by collecting initial erroneous thought chains (CoT) inference from VLM, and then using VLM self reflection to generate corrected CoT trajectories. During inference, RAG retrieves these corrected examples as minority shot examples to guide VLM's inference. This article also contributes a new manual annotation CAD dataset and a two-stage pipeline to solve the problem of long context. Experiments have shown that GPT-4o has improved accuracy by 23.4% on a human preference benchmark

**Strengths:**

1. The authors contribute a new multimodal CAD dataset that includes human preference annotations. This would be  a valuable resource for the community if it can be made public.

2. The concept of leveraging corrected reasoning exemplars as reusable Error Notebooks is novel and elegant.

3. This work provides a new approach for solving the problem of excessively long STEP file inputs, and the part retrieval in assemblies that it focuses on is also a broad application in CAD design.

**Weaknesses:**

1. Some typos need further correction, such as missing commas after some equations and missing spaces between some words.

2. The latency and resource overhead of RAG-based prompting are not quantified (e.g. the time for doing one complete inference).

3. The core viewpoint is that retrieving corrected error is crucial. However, the experiment (Table 1) only compared "w/E-Notebook" (its method) with "w/o-E-Notebook" (zero sample). Missing a key baseline: a standard few shot method based on RAG that only retrieves correct examples (i.e. (query, correct_CoT, correct_answer)). Without this comparison, it is hard to know whether the performance gain comes from  error corrections or solely from known benefits provided by RAG with any relevant contextual examples.

**Questions:**

1. This paper notes that CoT performance drops with 50 exemplars due to long prompt lengths. Does this suggest a fundamental limitation? Have authors explored strategies to mitigate this, such as summarizing the retrieved CoTs or dynamically selecting only the most relevant steps from each CoT?

2. What is the average token length and inference time per query? This is quite important for practical applications, as it is difficult to apply if the output result is too slow due to the long time of doing CoT.

3. Could authors please quantify the computational cost of building the Error Notebook? For example, how many VLM calls are required per sample to generate the corrected CoT?

4. Are the geometric constraints required for part assembly essentially obtained from the retrieved STEP files, rather than being calculated by the LLM/VLM directly?

---

> ### Author Response · Authors · 2025-11-20
>
> Dear Reviewer Hfy1,
>
> Thank you very much for your encouraging comments. **Sincerely appreciate** you highlighted our concept of leveraging corrected reasoning exemplars as reusable Error Notebooks, and the contribution of our new human-annotated multimodal CAD dataset. As you kindly noted, this dataset would be a valuable resource for the community if made public. **To better benefit the AI community, we have updated and provided the human-in-the-loop CAD dataset (see *dataset* in the updated Supplementary Material)**.
>
> Below is our point-by-point response to your comments (both Weaknesses and Questions):
>
> ***
>
> ### **[For the Weaknesses section]**
>
> > **W1: Some typos need further correction, such as missing commas after some equations and missing spaces between some words.**
>
> W1Re: Thank you very much for pointing this out. We carefully reviewed our manuscript and **fixed missing commas, and missing spaces**. Furthermore, **inspired by your comment, we assigned equation numbers to improve readability**. The revisions include:
>  - Line 177: added missing comma and equation number
>  - Line 186: added missing comma and equation number
>  - Line 210: added missing comma and equation number
>  - Line 214: added equation number
>  - Line 262: added missing comma and equation number
>  - Line 295: added missing comma and equation number
>  - Line 254: corrected missing space (*theError Notebook* -> *the Error Notebook*)
>
> > **W2: The latency and resource overhead of RAG-based prompting are not quantified (e.g. the time for doing one complete inference).**
>
> W2Re: Thank you for pointing this out. We conducted a runtime and token-cost evaluation on **100 samples** under: **GPT-4o (Omni)**, **single endpoint**, `max_workers=1`, no batching.
>
> | Setting | Avg Time (s) | Prompt Tokens | Completion Tokens |
> |-|-|-|-|
> | 1st VLM (part description) | 78.32 | — | — |
> | **2nd VLM (w/o Error Notebook)** | 8.04 | 967.7 | 235.4 |
> | **2nd VLM (w/ Error Notebook)** | 6.50 | 1815.3 | 278.7 |
> | **CoT Correction Step** | 7.39 | 1328.7 | 377.5 |
>
> **Therefore,**
>
> - Although using Error Notebook increases prompt tokens, **inference does not become slower** (8.04s vs 6.50s). Corrected exemplars may improve reasoning coherence and reduce internal search depth.
> - The one-time correction step is lightweight (**7.39 s per sample**).
> - 1st VLM latency is high since it depends on the **number** of STEP parts.
>
> Overall, **RAG-based prompting does not introduce prohibitive overhead**.
>
> > **W3: The core viewpoint is that retrieving corrected error is crucial. However, the experiment (Table 1) only compared "w/E-Notebook" (its method) with "w/o-E-Notebook" (zero sample). Missing a key baseline: a standard few shot method based on RAG that only retrieves correct examples (i.e. (query, correct_CoT, correct_answer)). Without this comparison, it is hard to know whether the performance gain comes from error corrections or solely from known benefits provided by RAG with any relevant contextual examples.**
>
> W3Re: Thank you very much for your feedback. Following your advice, we ran **new experiments** to compare our method against the standard few-shot learning. The settings are as follows: GPT-4o (Omni) with 2 API endpoints, and adopt two GPT-generated exemplars as few-shot examples (the number of exemplars is aligned with the **2-exemplar** setting in Table 1). **Those GPT-generated exemplars have the formats: query, correct_CoT, correct_answer**. We keep the full two-stage pipeline: the 1st VLM generates part descriptions from the assembly and part images; the second VLM performs reasoning. Standard few-shot prompting is applied to the second VLM (reasoning stage).
>
> *SG=Self-generated dataset, HP=Human preference dataset*
>
> | | SG: Overall | <10 | 10–20 | 20–50 | >50 | HP: Overall | <10 | 10–20 | 20–50 | >50 |
> |-|-|-|-|-|-|-|-|-|-|-|
> | **standard few-shot** | 26.6 | 37.4 | 19.2 | 16.9 | 6.2 | 37.7 | 42.9 | 29.4 | 17.6 | 21.4 |
> | **w/o E-Notebook** | 28.5 | 40.7 | 22.4 | 15.3 | 5.0 | 41.7 | 47.9 | 32.4 | 26.5 | 0.0 |
> | **self-consistency** | 38.9 | 54.6 | 30.1 | 21.2 | 11.3 | 54.8 | 61.7 | 42.6 | 29.4 | 35.7 |
> | **w/ E-Notebook (ours)** | **48.3** | **66.8** | **35.9** | **29.7** | **16.3** | **65.1** | **75.5** | **42.6** | **41.2** | **21.4** |
>
> **Conclusion: Our method outperforms existing baselines: standard few-shot across all groups.** We have added these results and analysis to the revised submission.

---

> ### Author Response · Authors · 2025-11-20
>
> ### **[For the Questions section]**
>
> > **Q1: This paper notes that CoT performance drops with 50 exemplars due to long prompt lengths. Does this suggest a fundamental limitation? Have authors explored strategies to mitigate this, such as summarizing the retrieved CoTs or dynamically selecting only the most relevant steps from each CoT?**
>
> Q1Re: Thank you for raising this very interesting question, and we agree that **summarizing retrieved CoTs or dynamically selecting only the most relevant reasoning steps** could mitigate the issue.
>
> However, we do not view this as a *fundamental limitation* of our framework, for several reasons:
>
> - **This mainly reflects the VLM’s own long-context limitations (e.g., GPT/Gemini’s handling of very long prompts)**, rather than a limitation of our method itself. Our framework does not train new models but enhances already-strong VLMs during inference. Therefore, improvements in model capacity or long-context handling would directly benefit the 50-exemplar case.
>
> - **Table 2 gives a practical solution**: performance consistently improves when going from 1 to 20 exemplars, and **20** exemplars strike a good balance between accuracy and prompt length.
>
> - **Long prompts naturally lead to slower inference**. From a deployment perspective, very long exemplars are not recommended regardless of the method.
>
> > **Q2: What is the average token length and inference time per query? This is quite important for practical applications, as it is difficult to apply if the output result is too slow due to the long time of doing CoT.**
>
> Q2Re: Thank you for the question. Please kindly refer to our response to Weakness #2.
>
> > **Q3: Could authors please quantify the computational cost of building the Error Notebook? For example, how many VLM calls are required per sample to generate the corrected CoT?**
>
> Q3Re: Based on our analysis, the total number of VLM calls required to construct the Error Notebook is:
>
> $$
> \sum_{i=1}^{n} (\text{part count}_i + 1) + n \times 1
> $$
>
> Here, *n* is the number of samples included in the Error Notebook.
>
> **The first term:** For each sample *i* with *part count_i* parts, we need *part count_i* VLM calls to generate the descriptions for all individual parts. Then, **one** additional VLM call is made to produce the initial CoT and its retrieval result, which explains the **+1** term inside the summation.
>
> **The second term:** each of the **n** samples must undergo **one** more VLM call to generate the corrected CoT, resulting in the **n x 1** term.
>
> Therefore, if we want to compute the **average** VLM calls across the data (**calls per sample**), we simply divide by *n*, giving:
>
> $$
> \frac{\sum_{i=1}^{n} (\text{part count}_i + 1) + n \times 1}{n}
> $$
>
> > **Q4: Are the geometric constraints required for part assembly essentially obtained from the retrieved STEP files, rather than being calculated by the LLM/VLM directly?**
>
> Q4Re: Thank you for this important question, and we apologize for the confusion. In fact, the geometric constraints required for part assembly are derived from **both** the retrieved STEP files and our two VLMs. To clarify this clearly, we have prepared a PDF titled **1_scope.pdf** in the updated Supplementary Material for your reference.
>
> Our framework takes two inputs:
>
> (1) **the assembly STEP file (in metadata form)**,
>
> (2) **the specification (in natural language)**.
>
> For (1), **geometric constraints/features of the assembly STEP file (in metadata form)**, these features originate from the STEP data but must still be processed by the 1st VLM. *A STEP file does not contain explicit natural-language descriptions of shapes,* only 3D metadata, so the system:
>  - first parses the assembly **STEP file**, splits it into individual part STEP files (via libraries/rules, not a VLM),
>  - converts them into part images (also via libraries/rules).
>  - the **1st VLM** then analyzes these part images and generates the geometric descriptions.
>
> For (2), **geometric constraints of the specification (natural language)**, the STEP files are no longer needed at this stage. Here, the 2nd VLM reasons purely from the natural-language specification and the part descriptions from the 1st VLM, and it computes the required constraints directly.
>
> **We appreciate this helpful comment and have incorporated these clarifications into the revised paper.**

---

> > ### Author Response · Authors · 2025-11-25
> >
> > Dear Reviewer Hfy1, :)
> >
> > Thank you again for your thoughtful review and constructive suggestions! Below we summarize the key contributions of our work as they appear in the updated version:
> >
> > (1) We propose a training-free reasoning framework that integrates the Error Notebook with RAG for VLM inference. **Importantly, the method outperforms traditional training-free inference-time approaches (standard few-shot, self-consistency) and also shows strong improvements on open-source models (e.g., Qwen2-VL-2B-Instruct and Aya-Vision-8B).**
> >
> > (2) We construct a new CAD assembly dataset with relational specifications and human-preference annotations. **As per your kind comments, the full human-annotated dataset is now released to support reproducibility and further research in this area.**
> >
> > (3) From an engineering standpoint, our two-stage pipeline effectively addresses the challenge of interpreting extremely long, non-natural-language STEP-file metadata.
> >
> > (4) The workflow for CoT correction is itself novel, and we provide a detailed illustration in the newly added 2_CoT.pdf in the updated Supplementary Materials.
> >
> > **If you have any additional questions or would find further clarification helpful, please feel free to reach out. We would be very glad to continue the discussion.**
> >
> > Thank you again for your time and consideration.
> >
> > Best regards,
> >
> > The Authors

---

> > > ### Comment · Reviewer_Hfy1 · 2025-11-26
> > >
> > > I thank the authors for their detailed response. The clarifications provided have addressed most of my concerns. Although some minor imperfections remain, they do not affect my initial assessment of the paper's contribution. I believe this is an interesting and worthwhile problem to explore in the field of CAD assembly. I would maintain my positive rating.

---

> > > > ### Author Response · Authors · 2025-11-26
> > > >
> > > > Dear Reviewer Hfy1,
> > > >
> > > > **Thank you so much for your positive rating!**
> > > >
> > > > ---
> > > >
> > > > Since in your earlier round you raised the insightful concern:
> > > >
> > > > > **The core viewpoint is that retrieving corrected error is crucial... it is hard to know whether the performance gain comes from error corrections or solely from known benefits provided by RAG with any relevant contextual examples.**
> > > >
> > > > We think that our previous response may not have fully addressed your concern, **so we revisited the retrieval process to further verify that it comes specifically from *high-quality* corrected errors rather than from any examples provided by RAG.**
> > > >
> > > > To do so, we defined HIGH-QUALITY corrected entries as those that pass a rule-based grammar constraint check. In particular, we verified whether each corrected CoT satisfies:
> > > >
> > > > ```
> > > > def passes_grammar_checks(cot_text, allowed_parts):
> > > >     m = re.search(r"Final Answer:\s*(.+)", cot_text)
> > > >     if not m:
> > > >         return False, "[NO_FINAL_ANSWER]"
> > > >     answer_str = m.group(1).strip()
> > > >     parts = [p.strip() for p in answer_str.split(";") if p.strip()]
> > > >     if not parts:
> > > >         return False, "[EMPTY_FINAL_ANSWER]"
> > > >     for p in parts:
> > > >         if p not in allowed_parts:
> > > >             return False, f"[INVALID_PART:{p}]"
> > > >     return True, "[OK]”
> > > > ```
> > > >
> > > > to ensure that the corrected CoT ends with a syntactically valid *Final Answer* containing only valid part filenames. We then rebuilt the Error Notebook using only entries that passed this grammar constraints (GC) check, and re-ran inference with the same RAG pipeline.
> > > >
> > > > *GC = grammar constraints*
> > > >
> > > > | | **SG: Overall** | <10 | 10–20 | 20–50 | >50 | **HP: Overall** | <10 | 10–20 | 20–50 | >50 |
> > > > |--------------|-----------------|------|--------|--------|------|-----------------|------|--------|--------|------|
> > > > | **GPT-4o (Omni)** |||||||||||
> > > > | w/o E-Notebook | 28.5 | 40.7 | 22.4 | 15.3 | 5.0 | 41.7 | 47.9 | 32.4 | 26.5 | 0.0 |
> > > > | w/ E-Notebook | 48.3 | 66.8 | 35.9 | 29.7 | **16.3** | 65.1 | 75.5 | 42.6 | 41.2 | 21.4 |
> > > > | w/ E-Notebook+GC | **48.5** | **67.0** | **36.5** | **32.2** | 12.5 | **66.8** | **75.5** | **48.5** | **50.0** | **21.4** |
> > > > | **GPT-4o mini** |||||||||||
> > > > | w/o E-Notebook | 13.6 | 20.5 | 10.9 | 4.2 | 1.3 | 19.3 | 24.8 | 10.3 | 0.0 | 0.0 |
> > > > | w/ E-Notebook | 24.9 | 34.9 | **25.0** | 5.9 | **7.5** | 35.4 | 41.5 | 29.4 | 8.8 | 7.1 |
> > > > | w/ E-Notebook+GC | **25.9** | **37.7** | 20.5 | **11.0** | 5.0 | **36.4** | **42.6** | **29.4** | **11.8** | **7.1** |
> > > > | **Gemini 1.5 Pro NS** |||||||||||
> > > > | w/o E-Notebook | 29.9 | 44.3 | 21.2 | 13.6 | 6.2 | 43.0 | 52.1 | 23.5 | 17.6 | 14.3 |
> > > > | w/ E-Notebook | 32.4 | 49.3 | 22.4 | 11.9 | 6.2 | 46.7 | 57.1 | 25.0 | **17.6** | 14.3 |
> > > > | w/ E-Notebook+GC | **36.2** | **51.8** | **26.3** | **17.8** | **12.5** | **50.3** | **60.6** | **30.9** | 14.7 | **21.4** |
> > > > | **Cloud Vision + Gemini 2.0 Flash** |||||||||||
> > > > | w/o E-Notebook | 35.0 | 51.8 | 25.0 | 14.4 | 8.7 | 50.0 | 58.9 | 38.2 | 17.6 | 7.1 |
> > > > | w/ E-Notebook | 40.4 | 59.3 | 30.1 | 16.1 | 11.3 | 57.8 | 66.3 | 47.1 | **29.4** | 7.1 |
> > > > | w/ E-Notebook+GC | **43.2** | **63.2** | **32.7** | **17.8** | **11.3** | **62.3** | **73.0** | **48.5** | 20.6 | **14.3** |
> > > > | **Gemini 2.5 Pro NS** |||||||||||
> > > > | w/o E-Notebook | 36.5 | 55.1 | 25.6 | 14.4 | 6.2 | 54.0 | 65.2 | 35.3 | 20.6 | 0.0 |
> > > > | w/ E-Notebook | 42.2 | 60.9 | **30.8** | **21.2** | **11.3** | 59.5 | 69.5 | **42.6** | 29.4 | **14.3** |
> > > > | w/ E-Notebook+GC | **42.9** | **64.8** | 28.8 | 20.3 | 5.0 | **62.1** | **74.1** | 38.2 | **32.4** | 7.1 |
> > > > | **Gemini 2.0 Flash NS** |||||||||||
> > > > | w/o E-Notebook | 30.9 | 46.8 | 21.2 | 12.7 | 5.0 | 44.2 | 53.5 | 23.5 | 20.6 | 14.3 |
> > > > | w/ E-Notebook | **40.4** | **58.2** | **31.4** | 19.5 | 8.7 | 56.8 | **67.0** | 39.7 | 23.5 | 14.3 |
> > > > | w/ E-Notebook+GC | 40.3 | 57.3 | 30.1 | **19.5** | **13.8** | **57.0** | 66.3 | **39.7** | **29.4** | **21.4** |
> > > >
> > > > This may respond better to your core concern: **the performance further improves via retrieving higher-quality corrected errors**. The improvement disappears if we relax these quality conditions.
> > > >
> > > > We sincerely appreciate your inspiring question and hope that this new evidence fully addresses your concern.
> > > >
> > > >
> > > > Best regards,
> > > >
> > > > The Authors

---

### Official Review · Reviewer_eRWS · 2025-10-31

**Soundness:** 3
**Presentation:** 2
**Contribution:** 2
**Rating:** 4
**Confidence:** 3

**Summary:**

This paper introduces a training-free framework for 3D CAD part retrieval using Vision-Language Models (VLMs). Its core contribution is the Error Notebook, a mechanism that uses Retrieval-Augmented Generation (RAG) to guide VLM inference with corrected reasoning exemplars, avoiding costly fine-tuning.

Key contributions: i) A training-free reasoning framework (Error Notebook + RAG) that significantly improves accuracy (up to 23.4% absolute gain) without fine-tuning. ii) A new, human-annotated CAD assembly dataset with relational specifications, based on the Fusion 360 Gallery. iii) A two-stage VLM pipeline (part description generation, then retrieval) to handle long, complex STEP file inputs.

**Strengths:**

The Error Notebook is a creative solution for improving proprietary, "black-box" VLMs that cannot be fine-tuned. It effectively adapts self-correction concepts to an inference-time strategy.

The evaluation is comprehensive, testing multiple VLMs (GPT-4o, Gemini) across various assembly complexities. The method shows robust and consistent improvements for all models, such as boosting GPT-4o (Omni) from 41.7% to 65.1% accuracy.

The two-stage pipeline is a practical engineering solution for processing lengthy, non-natural language STEP file metadata, making the approach viable for real CAD workflows.

The new human-preference dataset is a significant contribution, enabling more human-centric evaluation by filtering out ambiguous cases with multiple correct answers.

The ablations provide valuable insights, such as CoT reasoning being most beneficial for complex assemblies (>10 parts). The finding that performance is similar for 1 to 50 exemplars is also a key practical takeaway.

**Weaknesses:**

The paper is empirically strong but lacks a theoretical justification for why the Error Notebook works. There is no formal analysis of its properties or potential failure conditions.

The approach requires ground-truth labels to build the Error Notebook. This reliance limits its generalizability and scalability to new domains where labels are unavailable or expensive to acquire.

The primary comparison is against the same models without the Error Notebook. The paper needs comparisons against other training-free methods (e.g., standard few-shot learning, self-consistency) to isolate the specific benefit of the Error Notebook.

The computational overhead (API costs, token usage, latency) of constructing the notebook and performing RAG-enhanced inference is not discussed.

**Questions:**

- What specific function is used for RAG retrieval? How sensitive are results to this choice?
- How can the Error Notebook be constructed or applied in settings where ground-truth labels are unavailable or expensive to obtain?
- How does the Error Notebook compare to other training-free methods like inference-time scaling (for example self-consistency)?
- What are the computational overheads (API calls, tokens, latency) for notebook construction and RAG-enhanced inference compared to the baseline?
- How often do the VLMs successfully generate valid reasoning corrections given the ground truth? What happens when the correction generation itself fails?

---

> ### Author Response · Authors · 2025-11-20
>
> Dear Reviewer eRWS,
>
> Thank you so much for all your positive comments on our work. Especially grateful that you highlighted the creativity of our *Error Notebook* as an inference-time reasoning framework. Also sincerely appreciate your recognition of our new human-preference CAD dataset. **To better benefit the AI community, we have updated and provided the human-in-the-loop CAD dataset (see *dataset* in the updated Supplementary Material)**.
>
> Below is our point-by-point response to your comments (both Weaknesses and Questions):
>
> ***
>
> ### **[For the Weaknesses section]**
>
> > **W1: The paper is empirically strong but lacks a theoretical justification for why the Error Notebook works. There is no formal analysis of its properties or potential failure conditions.**
>
> W1Re: Thank you for raising this important point. Below we provide a **theoretical justification for why the Error Notebook works** and an **analysis of potential failure conditions**, supported by new experiments.
>
> **[Theoretical justification for why the Error Notebook works]**
>
> **1. Inference-time fine-tuning.** Although no training is performed, conditioning the model on *corrected CoT exemplars based on query similarity* effectively performs a form of **inference-time fine-tuning**. These exemplars reshape the model’s reasoning trajectory by:
> - constraining its solution space,
> - injecting task-specific reasoning priors,
> - reducing variance and stabilizing reasoning.
>
> **2. Large VLMs already contain the correct capability.** GPT-4o/Gemini can produce the correct answer by multiple attempts (**as supported by the self-consistency experiments in Weakness #3**), but their reasoning is unstable on complex inputs. The Error Notebook works by **suppressing spurious reasoning paths** and guiding the model toward its existing correct capability for much higher consistency.
>
> **[Analysis of potential failure conditions]**
>
> **Potential failure condition #1: small-sized models.** We added new experiments using the open-source model **Qwen2-VL-2B-Instruct**:
>
> | | SG: Overall | <10 | 10–20 | HP: Overall | <10 |
> |-|-|-|-|-|-|
> | **w/o E-Notebook** | 6/715 (0.8%) | 6 | 0 | 6/398 (1.5%) | 6 |
> | **w/ E-Notebook** | 46/715 (6.4%) | 45 | 1 | 43/398 (10.8%) | 43 |
> | **Improvement** | +40 (6.7×) | +39 | +1 | +37 (7.2×) | +37 |
>
> *SG=Self-generated dataset, HP=Human preference dataset*
>
> Therefore,
> - When model capacity is very limited, the accuracy remains low.
> - However, Error Notebook + RAG **still significantly** improves performance.
>
> **Potential failure condition #2: the Error Notebook must contain correct CoT (incorrect CoT directly harms performance).** During the early development of our idea, we found if exemplars contain incorrect CoT, even when the prompt explicitly states the mistake reason, accuracy does not improve:
>
> | Experiment ID | 1 | 2 | 3 | 4 | 5 | Test |
> |-|-|-|-|-|-|-|
> | Accuracy    | 466/726 | 456/726 | 463/726 | 457/726 | 465/726 | 441/726 |
>
> The first five runs correspond to attempts without an Error Notebook. The last *Test* run uses the incorrect CoTs from the previous attempts as RAG exemplars, after prompting the model with the explanation, but no improvement is observed (even drops to 441/726). This supports our design choice:
> - The Error Notebook **cannot** include erroneous CoT (even if error messages or explanations are added).
> - Hence, we introduced a dedicated **CoT correction step** to ensure the Error Notebook contains **only corrected reasoning**.
>
> > **W2: The approach requires ground-truth labels to build the Error Notebook. This reliance limits its generalizability and scalability to new domains where labels are unavailable or expensive to acquire.**
>
> W2Re: Thank you for pointing this out. Yes, and we agree that the need for ground-truth labels is a limitation, and we will update the *Limitations* section accordingly.
>
> However, our experiments (see Weakness #3) consistently show that our method remains more stable and significantly more accurate than **fully training-free, ground-truth label-free baselines**.

---

> > ### Author Response · Authors · 2025-11-20
> >
> > > **W3: The primary comparison is against the same models without the Error Notebook. The paper needs comparisons against other training-free methods (e.g., standard few-shot learning, self-consistency) to isolate the specific benefit of the Error Notebook.**
> >
> > W3Re: Thank you! Following your advice, we ran **new experiments** to compare our method against: standard few-shot learning, self-consistency. The settings are as follows:
> >
> > **Standard few-shot learning.** GPT-4o (Omni) with 2 API endpoints, and adopt two GPT-generated exemplars as few-shot examples (the number of exemplars is aligned with Table 1). We keep the full two-stage pipeline: the 1st VLM generates part descriptions from the assembly and part images; the second VLM performs reasoning. Standard few-shot prompting is applied to the second VLM (reasoning stage).
> >
> > **Self-consistency.** We keep the same two-stage VLM pipeline as in our main method. The first VLM generates part-level descriptions exactly as before. For the second VLM, we remove Error Notebook guidance, and only use GPT-4o (Omni) with 2 API endpoints and set temperature = 0.7. For each query, we call the second-stage VLM 5 times. We then perform majority voting over these results.
> >
> > *SG=Self-generated dataset, HP=Human preference dataset*
> >
> > | | SG: Overall | <10 | 10–20 | 20–50 | >50 | HP: Overall | <10 | 10–20 | 20–50 | >50 |
> > |-|-|-|-|-|-|-|-|-|-|-|
> > | **standard few-shot** | 26.6 | 37.4 | 19.2 | 16.9 | 6.2 | 37.7 | 42.9 | 29.4 | 17.6 | 21.4 |
> > | **w/o E-Notebook** | 28.5 | 40.7 | 22.4 | 15.3 | 5.0 | 41.7 | 47.9 | 32.4 | 26.5 | 0.0 |
> > | **self-consistency** | 38.9 | 54.6 | 30.1 | 21.2 | 11.3 | 54.8 | 61.7 | 42.6 | 29.4 | 35.7 |
> > | **w/ E-Notebook (ours)** | **48.3** | **66.8** | **35.9** | **29.7** | **16.3** | **65.1** | **75.5** | **42.6** | **41.2** | **21.4** |
> >
> > **Conclusion: our method outperforms existing baselines: standard few-shot, self-consistency, and the zero-shot w/o E-Notebook baseline across all groups.** We have added these results and analysis to the revised submission.
> >
> > > **W4: The computational overhead (API costs, token usage, latency) of constructing the notebook and performing RAG-enhanced inference is not discussed.**
> >
> > W4Re: Thank you for raising this point. We address this in 2 aspects:
> >
> > **1. Token Usage & Latency Overhead**
> >
> > We conducted a runtime and token-cost evaluation on **100 samples** under: **GPT-4o (Omni)**, **single endpoint**, `max_workers=1`, no batching.
> >
> > | Setting | Avg Time (s) | Prompt Tokens | Completion Tokens |
> > |-|-|-|-|
> > | 1st VLM (part description) | 78.32 | — | — |
> > | **2nd VLM (w/o Error Notebook)** | 8.04 | 967.7 | 235.4 |
> > | **2nd VLM (w/ Error Notebook)** | 6.50 | 1815.3 | 278.7 |
> > | **CoT Correction Step (Building Error Notebook)** | 7.39 | 1328.7 | 377.5 |
> >
> > **Therefore,**
> >
> > - Although using Error Notebook increases prompt tokens, **inference does not become slower** (8.04s vs 6.50s). Corrected exemplars may improve reasoning coherence and reduce internal search depth.
> > - The one-time correction step is lightweight (**7.39 s per sample**).
> > - 1st VLM latency is high since it depends on the **number** of STEP parts.
> >
> > Overall, **RAG-enhanced inference does not introduce prohibitive overhead**.
> >
> > **2. API Call Cost (Number of VLM Calls Required)**
> >
> > The total number of VLM calls needed to construct the Error Notebook is:
> >
> > $$
> > \sum_{i=1}^{n} (\text{part count}_i + 1) + n \times 1
> > $$
> >
> > Here, *n* is the number of samples included in the Error Notebook.
> >
> > **The first term:** For each sample *i* with *part count_i* parts, we need *part count_i* VLM calls to generate the descriptions for all individual parts. Then, **one** additional VLM call is made to produce the initial CoT and its retrieval result, which explains the **+1** term inside the summation.
> >
> > **The second term:** each of the **n** samples must undergo **one** more VLM call to generate the corrected CoT, resulting in the **n x 1** term.

---

> ### Author Response · Authors · 2025-11-20
>
> ### **[For the Questions section]**
>
> > **Q1: What specific function is used for RAG retrieval? How sensitive are results to this choice?**
>
> Q1Re: Thank you for the question. In Table 1, our Error Notebook uses a **character-level similarity retriever**:
>
> ```
> SequenceMatcher().ratio()
> ```
>
> This computes a normalized character-level matching score between specifications.
>
> To answer *how sensitive are the results to this choice*, we conducted a group of new experiments and implemented an improved retriever based on **token-level Jaccard similarity**:
>
> ```
> def _jaccard_similarity(a, b):
>     ta = token_set(a)
>     tb = token_set(b)
>     return len(ta & tb) / len(ta | tb)
> ```
>
> This version **tokenizes** specifications and measures the overlap between token sets.
>
> Below we report the comparison:
>
> *SG=Self-generated dataset, HP=Human preference dataset*
>
> | Strategy | SG: Overall | <10 | 10–20 | 20–50 | >50 | HP: Overall | <10 | 10–20 | 20–50 | >50 |
> |----------|-------------------------|------|--------|--------|------|----------------------|------|--------|--------|------|
> | w/o E-Notebook (Table 1) | 28.5 | 40.7 | 22.4 | 15.3 | 5.0 | 41.7 | 47.9 | 32.4 | 26.5 | 0.0 |
> | **w/ E-Notebook (Table 1, character-level)** | **48.3** | 66.8 | 35.9 | 29.7 | 16.3 | **65.1** | 75.5 | 42.6 | 41.2 | 21.4 |
> | **w/ E-Notebook (New, token-level)** | **50.2** | 68.4 | 39.7 | 31.4 | 16.3 | **68.1** | 77.3 | 50.0 | 47.1 | 21.4 |
>
> Therefore,
> - The token-level Jaccard retriever produces **slightly higher accuracy (about +2% accuracy on the self-generated dataset)** across most part-count groups.
> - However, regardless of retriever choice, **the Error Notebook methods** consistently yield large and robust gains over the baseline w/o E-Notebook.
> - These results show that the method is not highly sensitive to the exact retrieval scoring function.
>
> > **Q2: How can the Error Notebook be constructed or applied in settings where ground-truth labels are unavailable or expensive to obtain?**
>
> Q2Re: We believe that **what the model truly needs** is the good-quality reasoning **pattern**, not the full distribution of all possible examples. In practice, the number of such patterns is small. Therefore, the amount of data needed can be limited. Also,
>
> - during **construction**: the Error Notebook can be expanded incrementally through **human-in-the-loop updates**. Alternatively, as shown in our paper, a strong VLM can be used to automatically produce corrected CoT traces.
>
> - during **application**: can be achieved presented in our paper.
>
> > **Q3: How does the Error Notebook compare to other training-free methods like inference-time scaling (for example self-consistency)?**
>
> Q3Re: Thank you! Kindly refer to our response to Weakness #3.
>
> > **Q4: What are the computational overheads (API calls, tokens, latency) for notebook construction and RAG-enhanced inference compared to the baseline?**
>
> Q4Re: Thank you for raising this point. Kindly refer to our response to Weakness #4.
>
> > **Q5: How often do the VLMs successfully generate valid reasoning corrections given the ground truth? What happens when the correction generation itself fails?**
>
> Q5Re: There are several cases where the VLMs generate invalid reasoning corrections given the gt, and we do observe scenarios where self-correction fails.
>
> When the correction generation itself fails, they can reduce the final accuracy. **However, this performance drop can be effectively mitigated by using more exemplars**, as shown below.
>
> **Evidence: 20 exemplars consistently outperform 1 exemplar.** The trend is consistent across both datasets.
>
> **Table 2 (assemblies with ≥10 parts)**
>
> | #Exemplars | 10–20 parts | 20–50 parts | >50 parts |
> |-|-|-|-|
> *Self-generated dataset (CoT Group)*
> | 1          | 39.7        | 30.5        | 12.5      |
> | 20         | **42.3**    | **35.6**    | **16.3**  |
> *Human-preference dataset (CoT Group)*
> | 1          | 54.4        | 38.2        | 7.1       |
> | 20         | **57.4**    | **52.9**    | **7.1**   |
>
> In summary, **using only 1 exemplar increases the likelihood of retrieving an erroneous CoT**, which limits performance. In contrast, **using more exemplars (20 vs. 1) mitigates these occasional correction failures**, because the model can rely on a broader pool of corrected trajectories.
>
> **We hope these clarifications help address the concerns and allow for a re-assessment of our contribution.**

---

> ### Comment · Reviewer_eRWS · 2025-11-21
>
> Thank you for the detailed response. Overall, I am satisfied with the rebuttal and the experiments. I find the paper and results interesting. Few notes:
>
> > Inference-time fine-tuning. Although no training is performed, conditioning the model on corrected CoT exemplars based on query similarity effectively performs a form of inference-time fine-tuning.
>
> I understand your point but I do not think this is the correct framing. You are conditioning the model in-context, not fine-tuning the learner. At most, I would call this approach inference-time adaptation or in-context adaptation. Notice that if you were doing inference-time fine-tuning, it should be possible to obtain a similar performance boost you report in Table 1 without the need for retrieval at inference-time.
>
> > small-sized models
>
> I find it interesting that the self-generated and human-preference dataset provide almost the same improvement with a small capacity model.
>
> > Hence, we introduced a dedicated CoT correction step to ensure the Error Notebook contains only corrected reasoning
>
> Maybe I missed this point: when you fix the Error Notebook, how do you verify/assure that the CoT traces are correct?

---

> ### Author Response · Authors · 2025-11-24
>
> Dear Reviewer eRWS, :)
>
> Thank you so much for your feedback! This has gotten us to reconsider some of the parts in our work. **We're delighted that, through new experiments,  your new insights have inspired new findings in our work**. Now we still respond to your *Notes* one by one:
>
> > **Note1: I understand your point but I do not think this is the correct framing. You are conditioning the model in-context, not fine-tuning the learner. At most, I would call this approach inference-time adaptation or in-context adaptation. Notice that if you were doing inference-time fine-tuning, it should be possible to obtain a similar performance boost you report in Table 1 without the need for retrieval at inference-time.**
>
> Note1Re: Thank you for this very valid point. After careful consideration, we agree that calling our approach *inference-time fine-tuning* is overstated. We will avoid using this term in the main paper. And to better benefit the AI community, we will remove the corresponding part from our rebuttal as well:
>
> ~~**1. Inference-time fine-tuning.** Although no training is performed, conditioning the model on *corrected CoT exemplars based on query similarity* effectively performs a form of **inference-time fine-tuning**. These exemplars reshape the model’s reasoning trajectory by:~~
>
> ~~- constraining its solution space,~~
>
> ~~- injecting task-specific reasoning priors,~~
>
> ~~- reducing variance and stabilizing reasoning.~~
>
> In the revised version of the paper, we adopt your suggested terminology **inference-time adaptation or in-context adaptation** as the definition of our method.
>
> Following your helpful feedback, now our contributions are as follows:
>
> (1) Propose an **inference-time (in-context) adaptation reasoning framework** combining the Error Notebook and RAG for VLM inference. **Importantly, our method surpasses traditional training-free inference-time approaches (standard few-shot, self-consistency) and even shows strong improvements on open-source models** (e.g., Qwen2-VL-2B-Instruct and **Aya-Vision-8B (new finding)**).
>
> (2) Construct a new multimodal CAD assembly dataset with relational specifications and human annotations. **And we have released this human-annotated dataset for reproducibility.**
>
> (3) From an engineering perspective, we design an effective two-stage VLM strategy that first generates part descriptions and then uses these descriptions for retrieval, overcoming the challenge of interpreting extremely long STEP-file metadata.
>
> (4) **The construction procedure of our Error Notebook is itself novel, as illustrated in the NEWLY ADDED 2_CoT.pdf in the updated Supplementary Materials.**
>
> We hope that this additional clarification presents our method in a more solid manner.
>
> > **Note2: I find it interesting that the self-generated and human-preference dataset provide almost the same improvement with a small capacity model.**
>
> Note2Re: Thank you very much for your positive comments regarding the results on small-sized models. Below we provide the experimental details: the experiments were run on 8× A40 GPUs over approximately 3 days, with all settings (except the model itself) identical to those in Table 1. **We have released the code in the updated Supplementary Materials, under the folder Rebuttal_code_qwen.** And it's encouraging to observe that the Error Notebook + RAG framework **still yields meaningful improvements**, even for a small-capacity model.
>
> It is also worth noting that our human-preference dataset is a filtered subset of the self-generated dataset (**rules shown in lines 157-161 of the updated paper**). Because the underlying data distribution is similar, the two datasets understandably exhibit similar improvement trends under small models.
>
> To further validate the generality of our method, we additionally conducted **another** small-scale evaluation on **Aya-Vision-8B**. For efficiency, we used **7× A40 GPUs for ~36 hours**, and an additional run on **3× H20 GPUs for ~12 hours**. All experimental settings (except the model itself) remained identical to those in Table 1. The results show that Error Notebook + RAG continues to provide **clear and meaningful improvements**. We summarize the key improvements in the table below.
>
> *SG = Self-generated dataset (666 cases), HP = Human-preference dataset (370 cases)*
>
> |  | SG: Overall | <10 | 10–20 | HP: Overall | <10 | 10–20 |
> |--|--|--|--|--|--|--|
> | **w/o E-Notebook** | 16 (2.4%) | 16 | 0 | 14 (3.8%) | 14 | 0 |
> | **w/ E-Notebook** | 54 (8.1%) | 53 | 1 | 51 (13.8%)| 50 | 1|
> | **Improvement** | **+38 (3.4×)** | +37 | +1 | **+37 (3.6×)** | +36 | +1 |
>
> Overall, these results further strengthen our confidence that our method can generalize across a broad range of models, **not only proprietary VLMs but also small open-source ones**.

---

> > ### Author Response · Authors · 2025-11-24
> >
> > > **Note3: Maybe I missed this point: when you fix the Error Notebook, how do you verify/assure that the CoT traces are correct?**
> >
> > Note3Re: Thank you for this insightful question. To address it thoroughly, we created a dedicated new figure that illustrates the correction pipeline clearly. Please refer to **`2_CoT.pdf`** in the updated Supplementary Materials.
> > In our method, after the initial reasoning, we prompt the **same reasoning VLM** (e.g., Gemini or GPT-4o) to:
> >
> > 1. identify the **first** incorrect step in the original CoT,
> > 2. insert an appropriate **transition phrase**, and
> > 3. continue the reasoning until it reaches the **GT** part identifier.
> >
> > Importantly, **we do not perform an additional, external verification step**.
> >
> > However, during correction:
> >
> > - the **GT part identifier is explicitly provided**,
> > - forcing the model to generate a corrected trajectory that:
> >   1. preserves the valid prefix of the original CoT,
> >   2. inserts a transition,
> >   3. produces a new reasoning chain that **must terminate at the GT**.
> >
> > Thus each corrected CoT has the structure:
> >
> > **`Original CoT up to first error + Transition + New reasoning path + GT`**
> >
> > When applying the Error Notebook, to avoid leakage, we adopt a **leave-one-out filtering** strategy: the current test query is *removed* from the Error Notebook before retrieval. This ensures that RAG never retrieves an exemplar containing the query being evaluated.
> >
> > Finally, as you kindly pointed out in Q5, VLMs occasionally produce invalid corrections even when the GT is provided. Our experiments show that this issue can be mitigated effectively by using **more exemplars** (e.g., 20 vs. 1).
> >
> > **If you have any additional questions, we would be very happy and open to continue the discussion! And we would greatly appreciate your reconsideration of our paper. :)**

---

> ### Comment · Reviewer_eRWS · 2025-11-24
>
> Thank you for the additional clarification and comments. I think in-context adaptation or inference-time adaptation frame your work much better.
>
> > It is also worth noting that our human-preference dataset is a filtered subset of the self-generated dataset (rules shown in lines 157-161 of the updated paper). Because the underlying data distribution is similar, the two datasets understandably exhibit similar improvement trends under small models.
>
> I find it interesting because, after the rule-based filtering, you are basically fine-tuning the model using rejection-sampling, that can be seen as a simple form of policy gradient with a rule-based reward. If the "reward/preference" provides useful feedback, tuning using your human-preference filtered set should clearly outperform the unfiltered synthetic dataset (think sampling a model and fine-tuning on all the samples vs sampling a model and selecting the best performing). The distributions should not be that similar.
>
> > Importantly, we do not perform an additional, external verification step.
>
> I see. I think it would be interesting to explore grammar constraints (similar to your rule-based filtering) for intermediate reasoning steps as a form of process reward for reasoning with CAD.
>
> Anyway, thanks for the clarification. I have raised my score.

---

> ### Author Response · Authors · 2025-11-25
>
> Dear Reviewer eRWS,
>
> Thank you so much for sharing this!
>
> > **...If the "reward/preference" provides useful feedback, tuning using your human-preference filtered set should clearly outperform the unfiltered synthetic dataset (think sampling a model and fine-tuning on all the samples vs sampling a model and selecting the best performing)...**
>
> Thank you and yes, this difference between the self-generated dataset and the human-preference dataset **is exactly what we expected**, as shown in Table 1, this pattern appears for large models.
>
> However, for small open-source models, this becomes less pronounced. We suspect this is because their base accuracy is low, so their behavior is no longer dominated by dataset-level preference signals. In this low-capacity regime, the *preference signal* becomes harder to manifest simply because the model struggles to produce stable trajectories at all.
>
> But your comment has truly inspired us, we now see a promising connection between **the rule-based preference filtering** and **rejection-sampling style preference optimization**, and we'd be excited to explore this direction further in future work.
>
> > **...I think it would be interesting to explore grammar constraints (similar to your rule-based filtering) for intermediate reasoning steps as a form of process reward for reasoning with CAD.**
>
> Thank you! We also agree this is a promising direction to further strengthen the *Error Notebook*, and will definitely explore this (now we feel that adding structured, rule-like consistency checks could help, like a form of **process-level reward shaping**), and we will incorporate this insightful point into the final paper.
>
> **Thank you again for this thoughtful and inspiring suggestion!**
>
> **Sincerely**,
>
> The Authors

---

> > ### Author Response · Authors · 2025-11-26
> >
> > Dear Reviewer eRWS,
> >
> > Thank you for your insightful feedback throughout the review process. Since in your earlier round you mentioned:
> >
> > > Maybe I missed this point: when you fix the Error Notebook, how do you verify/assure that the CoT traces are correct?
> >
> > and later suggested:
> >
> > > I think it would be interesting to explore grammar constraints (similar to your rule-based filtering) for intermediate reasoning steps as a form of process reward for reasoning with CAD.
> >
> > **Therefore, we tested the grammar constraints (rule-based filtering) for filtering Error Notebook entries.** We added the following check:
> >
> > ```
> > def passes_grammar_checks(cot_text, allowed_parts):
> >     m = re.search(r"Final Answer:\s*(.+)", cot_text)
> >     if not m:
> >         return False, "[NO_FINAL_ANSWER]"
> >     answer_str = m.group(1).strip()
> >     parts = [p.strip() for p in answer_str.split(";") if p.strip()]
> >     if not parts:
> >         return False, "[EMPTY_FINAL_ANSWER]"
> >     for p in parts:
> >         if p not in allowed_parts:
> >             return False, f"[INVALID_PART:{p}]"
> >     return True, "[OK]”
> > ```
> >
> > **to ensure that the corrected CoT ends with a syntactically valid *Final Answer* containing only valid part filenames.** We then rebuilt the Error Notebook using only entries that passed this grammar constraints (GC) check, and re-ran inference with the same RAG pipeline. **And this trick produces clear improvements, especially on the human-preference dataset.**
> >
> > *GC = grammar constraints*
> >
> > | | **SG: Overall** | <10 | 10–20 | 20–50 | >50 | **HP: Overall** | <10 | 10–20 | 20–50 | >50 |
> > |--------------|-----------------|------|--------|--------|------|-----------------|------|--------|--------|------|
> > | **GPT-4o (Omni)** |||||||||||
> > | w/o E-Notebook | 28.5 | 40.7 | 22.4 | 15.3 | 5.0 | 41.7 | 47.9 | 32.4 | 26.5 | 0.0 |
> > | w/ E-Notebook | 48.3 | 66.8 | 35.9 | 29.7 | **16.3** | 65.1 | 75.5 | 42.6 | 41.2 | 21.4 |
> > | w/ E-Notebook+GC | **48.5** | **67.0** | **36.5** | **32.2** | 12.5 | **66.8** | **75.5** | **48.5** | **50.0** | **21.4** |
> > | **GPT-4o mini** |||||||||||
> > | w/o E-Notebook | 13.6 | 20.5 | 10.9 | 4.2 | 1.3 | 19.3 | 24.8 | 10.3 | 0.0 | 0.0 |
> > | w/ E-Notebook | 24.9 | 34.9 | **25.0** | 5.9 | **7.5** | 35.4 | 41.5 | 29.4 | 8.8 | 7.1 |
> > | w/ E-Notebook+GC | **25.9** | **37.7** | 20.5 | **11.0** | 5.0 | **36.4** | **42.6** | **29.4** | **11.8** | **7.1** |
> > | **Gemini 1.5 Pro NS** |||||||||||
> > | w/o E-Notebook | 29.9 | 44.3 | 21.2 | 13.6 | 6.2 | 43.0 | 52.1 | 23.5 | 17.6 | 14.3 |
> > | w/ E-Notebook | 32.4 | 49.3 | 22.4 | 11.9 | 6.2 | 46.7 | 57.1 | 25.0 | **17.6** | 14.3 |
> > | w/ E-Notebook+GC | **36.2** | **51.8** | **26.3** | **17.8** | **12.5** | **50.3** | **60.6** | **30.9** | 14.7 | **21.4** |
> > | **Cloud Vision + Gemini 2.0 Flash** |||||||||||
> > | w/o E-Notebook | 35.0 | 51.8 | 25.0 | 14.4 | 8.7 | 50.0 | 58.9 | 38.2 | 17.6 | 7.1 |
> > | w/ E-Notebook | 40.4 | 59.3 | 30.1 | 16.1 | 11.3 | 57.8 | 66.3 | 47.1 | **29.4** | 7.1 |
> > | w/ E-Notebook+GC | **43.2** | **63.2** | **32.7** | **17.8** | **11.3** | **62.3** | **73.0** | **48.5** | 20.6 | **14.3** |
> > | **Gemini 2.5 Pro NS** |||||||||||
> > | w/o E-Notebook | 36.5 | 55.1 | 25.6 | 14.4 | 6.2 | 54.0 | 65.2 | 35.3 | 20.6 | 0.0 |
> > | w/ E-Notebook | 42.2 | 60.9 | **30.8** | **21.2** | **11.3** | 59.5 | 69.5 | **42.6** | 29.4 | **14.3** |
> > | w/ E-Notebook+GC | **42.9** | **64.8** | 28.8 | 20.3 | 5.0 | **62.1** | **74.1** | 38.2 | **32.4** | 7.1 |
> > | **Gemini 2.0 Flash NS** |||||||||||
> > | w/o E-Notebook | 30.9 | 46.8 | 21.2 | 12.7 | 5.0 | 44.2 | 53.5 | 23.5 | 20.6 | 14.3 |
> > | w/ E-Notebook | **40.4** | **58.2** | **31.4** | 19.5 | 8.7 | 56.8 | **67.0** | 39.7 | 23.5 | 14.3 |
> > | w/ E-Notebook+GC | 40.3 | 57.3 | 30.1 | **19.5** | **13.8** | **57.0** | 66.3 | **39.7** | **29.4** | **21.4** |
> >
> > So **this grammar-based filtering step leads to a clear improvement**, and we will include these updated results in the final version of the paper.
> >
> > We appreciate your thoughtful feedback and hope that the new evidence addresses your concerns.
> >
> > Best regards,
> >
> > The Authors

---

> > > ### Comment · Reviewer_eRWS · 2025-11-26
> > >
> > > Thank you for the additional results. Please add them to the paper and, if possible, run the same w/ E-Notebook+GC with the small models.

---

> > > > ### Author Response · Authors · 2025-11-27
> > > >
> > > > Dear eRWS,
> > > >
> > > > Thank you! Following your suggestion, we run the w/ E-Notebook+GC with Qwen2-VL-2B-Instruct. The results reveal several notable findings:
> > > >
> > > > *SG: 715 cases; <10: 361; 10-20: 156. HP: 398 cases; <10: 282*
> > > >
> > > > | | SG Overall | <10 | 10–20 | HP Overall | <10 |
> > > > |---------|------------|--------|----------|------------|--------|
> > > > | w/o E-Notebook | 0.8 (6) | 1.7 (6) | 0.0 (0) | 1.5 (6) | 2.1 (6) |
> > > > | w/ E-Notebook | 6.4 (46) | 12.5 (45) | 0.6 (1) | 10.8 (43) | 15.2 (43) |
> > > > | *Improvement* | +5.6 (+40) | +10.8 (+39) | +0.6 (+1) | +9.3 (+37) | +13.1 (+37) |
> > > > | **w/ E-Notebook + GC1** | 3.6 (26) | 7.2 (26) | 0.0 (0) | 6.0 (24) | 8.5 (24) |
> > > > | **w/ E-Notebook + GC2** | 6.6 (47) | 12.7 (46) | 0.6 (1) | 10.8 (43) | 15.2 (43) |
> > > > | **w/ E-Notebook + GC3 (cross-model)** | **8.4 (60)** | **16.6 (60)** | **0.0 (0)** | **14.6 (58)** | **20.6 (58)** |
> > > > | *Improvement (GC3 – w/o EN)* | +7.6 (+54) | +14.9 (+54) | +0.0 (+0) | +13.1 (+52) | +18.5 (+52) |
> > > >
> > > > As noted previously:
> > > >
> > > > ```
> > > > def passes_grammar_checks(cot_text, allowed_parts):
> > > >     m = re.search(r"Final Answer:\s*(.+)", cot_text)
> > > >     if not m:
> > > >         return False, "[NO_FINAL_ANSWER]"
> > > >     answer_str = m.group(1).strip()
> > > >     parts = [p.strip() for p in answer_str.split(";") if p.strip()]
> > > >     if not parts:
> > > >         return False, "[EMPTY_FINAL_ANSWER]"
> > > >     for p in parts:
> > > >         if p not in allowed_parts:
> > > >             return False, f"[INVALID_PART:{p}]"
> > > >     return True, "[OK]”
> > > > ```
> > > >
> > > > **(1) w/ E-Notebook + GC1:** GC1 applies the same filtering rule used for proprietary models:
> > > >
> > > > `r.get("CoT_Grammar_Status", "") == "[OK]"`
> > > >
> > > > However, we observe the model frequently produces **acceptable** CoTs that, **however**, do not explicitly include the *Final Answer:* key phrase. **This significantly *shrinks* the valid Error Notebook and leads to degraded performance.**
> > > >
> > > > **(2) w/ E-Notebook + GC2:** Therefore, we relaxed the filter to:
> > > >
> > > > `r.get("CoT_Grammar_Status", "") in {"[OK]", "[NO_FINAL_ANSWER]"}`
> > > >
> > > > This admits more plausible traces and improves accuracy **compared to the baseline: w/ E-Notebook**.
> > > >
> > > > **(3) w/ E-Notebook + GC3 (cross-model):** We further replaced the **`Qwen2-VL-2B-Instruct-generated Error Notebook`** with **`GPT 4o (Omni) -Instruct-generated Error Notebook`**, while still using Qwen2-VL-2B-Instruct for inference. And now GC3 applies the same filtering rule used for proprietary models:
> > > >
> > > > `r.get("CoT_Grammar_Status", "") == "[OK]"`
> > > >
> > > > Under this **cross-model distillation setting**, the model achieves the best results:
> > > >
> > > > - **On human preference dataset <10, this 2B model is only 4.2 points below the proprietary model GPT-4o mini**.
> > > > - The absolute overall accuracy of this 2B model on HP is even higher than GPT-4o mini on SG.
> > > >
> > > > We can now even say that **our Error Notebook approach enables a 2B model to perform only about 4 points below GPT-4o mini**. We will consider incorporating all these findings into the revised paper (or **gladly take any further suggestions you may have**).
> > > >
> > > > Thank you again for the highly constructive guidance.
> > > >
> > > > Sincerely,
> > > > **The Authors**

---

> > > > > ### Comment · Reviewer_eRWS · 2025-11-27
> > > > >
> > > > > Thank you for the additional experiments. Very informative. One minor point:
> > > > >
> > > > > > We can now even say that our Error Notebook approach enables a 2B model to perform only about 4 points below GPT-4o mini.
> > > > >
> > > > > For the w/ E-Notebook + GC3 version, you are using GPT4o (omni) as a "verifier". So, even if it is technically correct that Qwen is the inference model, I would not say that "the 2B model is performing only 4 points below GPT4o (mini)".

---

> > > > > > ### Author Response · Authors · 2025-11-28
> > > > > >
> > > > > > Dear eRWS,
> > > > > >
> > > > > > Thank you very much for this helpful clarification. We think a more accurate description is:
> > > > > >
> > > > > > > Under the cross-model GC setting, where the Error Notebook is constructed using GPT-4o (Omni) but inference is performed by the 2B model, the resulting system achieves performance that comes within roughly 4 points of GPT-4o mini.
> > > > > >
> > > > > > This can be viewed as a training-free **transfer of high-quality reasoning traces from a proprietary model to a 2B one.** We have updated the manuscript accordingly:
> > > > > > - **Lines ~254-270 (main)**: added a new subsection and the description of GC mechanisms (w/ updated definitions of GC1 / GC2 / GC3).
> > > > > > - **Lines ~378-410 (main)**: added the GC experiments for proprietary models, Table 1.
> > > > > > - **Lines ~476-515 (main)**: added the GC experiments for small models, **Table 4**.
> > > > > >
> > > > > > Please find our updated paper for your reference.**In this revised version of the paper, and following your suggestions, we have made an effort to ensure more precise and accurate method description.**
> > > > > >
> > > > > > **A special thank you again for your constructive feedback**. please feel free to raise any further questions.
> > > > > >
> > > > > >
> > > > > > Sincerely,
> > > > > >
> > > > > > The Authors

---

### Official Review · Reviewer_f2Wn · 2025-11-01

**Soundness:** 3
**Presentation:** 3
**Contribution:** 2
**Rating:** 4
**Confidence:** 4

**Summary:**

The authors propose a part retrieval framework that uses error notebooks + RAG for refined prompt engineering. Furthermore, they contribute a human-in-the-loop CAD dataset for evaluating the proposed approach. Evaluations are performed on closed source models and achieve upto a 23% accuracy improvement on human-preference benchmarks.

**Strengths:**

- error notebooks idea is a creative solution for test-time reasoning augmentation
- contribution of human-in-the-loop dataset

**Weaknesses:**

- comparisons are primarily on proprietary models. It would be great to see them on open-source models as well.
- the eval metrics are somewhat under-developed. For example, there is not analysis on retrieval relevance.
- baseline comparisons against conventional CAD part retrieval methods are missing

**Questions:**

1. why was the dataset size limited to using a single set (752 assemblies) of Fusion 360 Assembly Dataset?
2. i might have missed this, but, how are the corrected reasoning trajectories verified?
3. are there scenarios where the model is not able to self-correct with prompting?
4. what kind of assemblies does the method struggle with?

---

> ### Author Response · Authors · 2025-11-20
>
> Dear Reviewer f2Wn,
>
> Really appreciate your recognition of the creativity of our Error Notebook and the value of our new human-in-the-loop CAD dataset! **To better benefit the AI community, we have updated and provided the human-in-the-loop CAD dataset (see *dataset* in the updated Supplementary Material)**.
>
> Below we provide point-by-point responses to your comments.
>
> ***
>
> ### **[For the Weaknesses section]**
>
> > **W1: comparisons are primarily on proprietary models. It would be great to see them on open-source models as well.**
>
> W1Re: Thank you for the suggestion.
>
> **Why open-source models were not included in the initial submission?** Our primary motivation was that **proprietary VLMs (e.g., GPT-4o, Gemini) do not provide model weights**, making training-based improvements impossible. Therefore, a **training-free method** that enhances their reliability is especially impactful. However, for open-source models, in principle, we can **fine-tune** them to improve performance.
>
> Despite this, we conducted new experiments on a lightweight open-source model *Qwen2-VL-2B-Instruct*.  **And it’s exciting to see that the Error Notebook + RAG framework brings large and meaningful improvements**.
>
> We used **8× A40 GPUs** and approximately **3 days** to run these experiments. All experimental settings (except the model itself) remained identical to those in Table 1. We report **the number of correctly solved cases** as follows:
>
> *SG=Self-generated dataset (715 cases), HP=Human preference dataset (398 cases)*
>
> | | SG: Overall | <10 | 10–20 | HP: Overall | <10 |
> |-|-|-|-|-|-|
> | **w/o E-Notebook** | 6 | 6 | 0 | 6 | 6 |
> | **w/ E-Notebook** | 46 | 45 | 1 | 43 | 43 |
> | **Improvement** | **+40 (6.7×)** | +39 | +1 | **+37 (7.2×)** | +37 |
>
> **Therefore, even for a 2B open-source VLM (the baseline accuracy is low), our method brings a *large and meaningful gain*.** We have updated the main paper accordingly.
>
> > **W2: the eval metrics are somewhat under-developed. For example, there is not analysis on retrieval relevance.**
>
> W2Re: Thank you for pointing this out. Following the reviewer’s suggestion, we additionally computed **retrieval relevance metrics**. Below we summarize the *retrieval relevance* definition used in our evaluation:
> - **TP:** `|GT ∩ Pred|`
> - **FP:** `|Pred − GT|`
> - **FN:** `|GT − Pred|`
>
> Then:
> - **Recall = TP / (TP + FN)**
> - **F1 = 2PR / (P + R)**
>
> We report both **global averaged R/F1**, and **per-group R/F1** based on assembly part count (`<10`, `10–20`, `20–50`, `>50`) on the self-generated dataset.
>
>
> | Strategy | **Global Recall** | **Global F1** | <10 (R/F1) | 10–20 (R/F1) | 20–50 (R/F1) | >50 (R/F1) |
> |----------|------------------|---------------|-------------|--------------|--------------|------------|
> **GPT-4o (Omni)**
> | w/o E-Notebook | 0.406 | 0.532 | 0.520 / 0.664 | 0.362 / 0.481 | 0.277 / 0.370 | 0.171 / 0.239 |
> | w/ E-Notebook  | 0.692 | 0.686 | 0.828 / 0.837 | 0.644 / 0.629 | 0.557 / 0.534 | 0.367 / 0.364 |
> **GPT-4o mini**
> | w/o E-Notebook | 0.261 | 0.385 | 0.344 / 0.494 | 0.218 / 0.325 | 0.179 / 0.269 | 0.089 / 0.144 |
> | w/ E-Notebook  | 0.500 | 0.523 | 0.619 / 0.675 | 0.500 / 0.504 | 0.289 / 0.288 | 0.272 / 0.275 |
> **Gemini 2.5 Pro (Non-streaming)**
> | w/o E-Notebook | 0.627 | 0.607 | 0.778 / 0.781 | 0.571 / 0.532 | 0.451 / 0.416 | 0.316 / 0.304 |
> | w/ E-Notebook  | 0.662 | 0.595 | 0.815 / 0.796 | 0.590 / 0.569 | 0.472 / 0.444 | 0.392 / 0.225 |
> **Gemini 2.0 Flash (Non-streaming)**
> | w/o E-Notebook | 0.552 | 0.573 | 0.681 / 0.728 | 0.529 / 0.531 | 0.400 / 0.392 | 0.241 / 0.254 |
> | w/ E-Notebook  | 0.630 | 0.628 | 0.777 / 0.784 | 0.583 / 0.584 | 0.468 / 0.446 | 0.297 / 0.296 |
> **Gemini 1.5 Pro (Non-streaming)**
> | w/o E-Notebook | 0.565 | 0.554 | 0.717 / 0.727 | 0.522 / 0.497 | 0.366 / 0.340 | 0.253 / 0.247 |
> | w/ E-Notebook  | 0.575 | 0.557 | 0.745 / 0.738 | 0.474 / 0.456 | 0.396 / 0.362 | 0.272 / 0.261 |
> **Cloud Vision + Gemini 2.0 Flash**
> | w/o E-Notebook | 0.617 | 0.604 | 0.750 / 0.776 | 0.583 / 0.553 | 0.438 / 0.398 | 0.342 / 0.318 |
> | w/ E-Notebook  | 0.636 | 0.622 | 0.788 / 0.794 | 0.577 / 0.562 | 0.447 / 0.412 | 0.342 / 0.326 |
>
> **Across all models and difficulty groups, the Error Notebook consistently yields clear and meaningful improvements in retrieval relevance**. These results have been included in the revised submission.

---

> > ### Author Response · Authors · 2025-11-20
> >
> > > **W3: baseline comparisons against conventional CAD part retrieval methods are missing**
> >
> > W3Re: Really appreciate the feedback! To the best of our knowledge, no prior work has addressed specification-aware **part retrieval** from long, non-natural-language STEP metadata. Since STEP metadata is highly complex for humans to interpret, VLMs are currently **the only plausible way** to process such information. **Our work is the first to bridge natural-language engineering specifications** with **STEP-based 3D part retrieval via VLM reasoning**.
> >
> > Based on your comments, we further include the following **conventional training-free baseline comparisons**:
> >
> > - **Existing baseline comparison: standard few-shot learning**
> > - **Existing baseline comparison: self-consistency**
> >
> > Below are the experimental results.
> >
> > ---
> >
> > *SG=Self-generated dataset, HP=Human preference dataset*
> >
> > | | SG: Overall | <10 | 10–20 | 20–50 | >50 | HP: Overall | <10 | 10–20 | 20–50 | >50 |
> > |-|-|-|-|-|-|-|-|-|-|-|
> > | **standard few-shot** | 26.6 | 37.4 | 19.2 | 16.9 | 6.2 | 37.7 | 42.9 | 29.4 | 17.6 | 21.4 |
> > | **w/o E-Notebook** | 28.5 | 40.7 | 22.4 | 15.3 | 5.0 | 41.7 | 47.9 | 32.4 | 26.5 | 0.0 |
> > | **self-consistency** | 38.9 | 54.6 | 30.1 | 21.2 | 11.3 | 54.8 | 61.7 | 42.6 | 29.4 | 35.7 |
> > | **w/ E-Notebook (ours)** | **48.3** | **66.8** | **35.9** | **29.7** | **16.3** | **65.1** | **75.5** | **42.6** | **41.2** | **21.4** |
> >
> > **Conclusion: Our method outperforms existing baselines: standard few-shot, self-consistency, and the zero-shot w/o E-Notebook baseline across all groups.**
> >
> > Thank you again for the helpful comments!
> >
> > ***
> >
> > ### **[For the Questions section:]**
> >
> > > **Q1: why was the dataset size limited to using a single set (752 assemblies) of Fusion 360 Assembly Dataset?**
> >
> > Q1Re: Thank you for the question, because:
> >
> > **1. Data quality and distribution across sets are highly similar in the Fusion 360 Assembly Dataset.** Each set contains comparable numbers of assemblies and identical file types. Therefore, using one set yields representative results.
> >
> > **2. Human annotation cost.** For each assembly, we require human annotators to label and filter.
> >
> > **3. API-call cost grows with assembly size.** Constructing our dataset requires API calls proportional to assembly size:
> >
> > $$
> > \sum_{i=1}^{n} (\text{part count}_i + 1)
> > $$
> >
> > Given that part counts range from **2 to 249**, and n = 752, expanding to multiple sets would significantly increase cost.
> >
> > **4. Our method is inference-only, so the entire set serves as a test set.** Since our framework does not require training, all 752 assemblies function as test data, and already a sufficient set for our task. Additionally, to avoid data leakage when evaluating the Error Notebook-based method, we adopt a **leave-one-out–style filtering**: the current test instance is excluded from the Error Notebook before retrieval, ensuring no contamination of evaluation data.
> >
> > > **Q2: i might have missed this, but, how are the corrected reasoning trajectories verified?**
> >
> > Q2Re: Thank you for this insightful question. In our method, after the initial part-retrieval reasoning, the same strong reasoning VLM (e.g., Gemini, GPT) is prompted to identify the *first* error in the existing CoT, insert an appropriate transition phrase, and then continue the reasoning toward the correct ground-truth part identifier.
> >
> > **We do not perform an additional external verification step**. However, during the correction process, the **GT (part identifier) is explicitly provided to the model**. This constrains the corrected trajectory to:
> > 1. preserve the valid portions of the original CoT,
> > 2. insert a transition phrase, and
> > 3. generate a new reasoning path that **must terminate at the GT**.
> >
> > Thus, although not separately verified, the corrected CoT is implicitly quality-controlled by requiring it to logically lead to the known ground truth.
> >
> > > **Q3. are there scenarios where the model is not able to self-correct with prompting?**
> >
> > Q3Re: Yes, we do observe scenarios where self-correction fails. **The VLM does not always succeed in generating fully valid corrected trajectories.**
> > When such cases occur, they can reduce the final accuracy. **However, this accuracy reduction can be effectively mitigated by using more exemplars**, as shown below:
> >
> > **20 exemplars consistently outperform 1 exemplar.** The trend is consistent across both datasets.
> >
> > **Table 2 (assemblies with ≥10 parts)**
> >
> > | #Exemplars | 10–20 parts | 20–50 parts | >50 parts |
> > |-|-|-|-|
> > *Self-generated dataset (CoT Group)*
> > | 1          | 39.7        | 30.5        | 12.5      |
> > | 20         | **42.3**    | **35.6**    | **16.3**  |
> > *Human-preference dataset (CoT Group)*
> > | 1          | 54.4        | 38.2        | 7.1       |
> > | 20         | **57.4**    | **52.9**    | **7.1**   |
> >
> > In summary, **using more exemplars (20 vs. 1) mitigates these occasional correction failures**, because the model can rely on a broader pool of corrected trajectories.

---

> > > ### Author Response · Authors · 2025-11-20
> > >
> > > > **Q4. what kind of assemblies does the method struggle with?**
> > >
> > > Q4Re: As shown in Table 1, our method struggles with *assemblies with more than 50 parts*. However, this difficulty largely stems from the inherent limitations of GPT/Gemini themselves rather than from the proposed Error Notebook framework. Our main focus is on the *improvement capability* enabled by the Error Notebook. **Even under these challenging conditions, our method achieves improvements in 8 out of 12 settings. No degradation is observed.**
> > >
> > > The potential reason for the lower absolute accuracy is these large assemblies often contain many visually and functionally similar components, making it difficult for the 1st VLM to generate sufficiently distinctive descriptions. Then the second-stage retrieval becomes more ambiguous, which leads to reduced accuracy. This issue can be mitigated by designing more sophisticated prompts for the 1st VLM.
> > >
> > > **We appreciate this helpful comment and have incorporated these clarifications into the revised paper.**

---

> ### Author Response · Authors · 2025-11-25
>
> Dear Reviewer f2Wn,
>
> Thank you very much for your thoughtful feedback. To further address your comment:
>
> > **W1: comparisons are primarily on proprietary models. It would be great to see them on open-source models as well.**
>
> we conducted a new experiment beyond Qwen2-VL-2B-Instruct, using **Aya-Vision-8B**. For computational efficiency, we used **7× A40 GPUs for ~36 hours**, followed by an additional run on **3× H20 GPUs for ~12 hours**. All experimental settings (except the model itself) were kept identical to those in Table 1. The results confirm that the Error Notebook + RAG framework continues to provide **clear and meaningful improvements**.
>
> *SG = Self-generated dataset (666 cases), HP = Human-preference dataset (370 cases)*
>
> |  | SG: Overall | <10 | 10–20 | HP: Overall | <10 | 10–20 |
> |--|--|--|--|--|--|--|
> | **w/o E-Notebook** | 16 (2.4%) | 16 | 0 | 14 (3.8%) | 14 | 0 |
> | **w/ E-Notebook** | 54 (8.1%) | 53 | 1 | 51 (13.8%) | 50 | 1 |
> | **Improvement** | **+38 (3.4×)** | +37 | +1 | **+37 (3.6×)** | +36 | +1 |
>
> These consistent improvements further strengthen our confidence that the proposed method generalizes well across a broader range of models, **not only proprietary VLMs, but also open-source models as well**.
>
> If you have any additional questions or would like to discuss further, we would be very happy to engage! Thank you again for your valuable insights and for helping us improve the work. :)
>
> Sincerely,
> The Authors

---

> ### Author Response · Authors · 2025-11-26
>
> Dear Reviewer f2Wn,
>
> Thank you for your insightful feedback! Since in your earlier round you mentioned:
>
> > i might have missed this, but, how are the corrected reasoning trajectories verified?
>
> > are there scenarios where the model is not able to self-correct with prompting?
>
> In the initial manuscript, we do not perform an additional external verification step. As per your questions, **we reconsidered these and tested the grammar constraints (rule-based filtering) for filtering Error Notebook entries to verify the corrected reasoning trajectories.** We added the following check:
>
> ```
> def passes_grammar_checks(cot_text, allowed_parts):
>     m = re.search(r"Final Answer:\s*(.+)", cot_text)
>     if not m:
>         return False, "[NO_FINAL_ANSWER]"
>     answer_str = m.group(1).strip()
>     parts = [p.strip() for p in answer_str.split(";") if p.strip()]
>     if not parts:
>         return False, "[EMPTY_FINAL_ANSWER]"
>     for p in parts:
>         if p not in allowed_parts:
>             return False, f"[INVALID_PART:{p}]"
>     return True, "[OK]”
> ```
>
> **to ensure that the corrected CoT ends with a syntactically valid *Final Answer* containing only valid part filenames.** We then rebuilt the Error Notebook using only entries that passed this grammar constraints (GC) check, and re-ran inference with the same RAG pipeline. **And this trick produces clear improvements, especially on the human-preference dataset.**
>
> *GC = grammar constraints*
>
> | | **SG: Overall** | <10 | 10–20 | 20–50 | >50 | **HP: Overall** | <10 | 10–20 | 20–50 | >50 |
> |--------------|-----------------|------|--------|--------|------|-----------------|------|--------|--------|------|
> | **GPT-4o (Omni)** |||||||||||
> | w/o E-Notebook | 28.5 | 40.7 | 22.4 | 15.3 | 5.0 | 41.7 | 47.9 | 32.4 | 26.5 | 0.0 |
> | w/ E-Notebook | 48.3 | 66.8 | 35.9 | 29.7 | **16.3** | 65.1 | 75.5 | 42.6 | 41.2 | 21.4 |
> | w/ E-Notebook+GC | **48.5** | **67.0** | **36.5** | **32.2** | 12.5 | **66.8** | **75.5** | **48.5** | **50.0** | **21.4** |
> | **GPT-4o mini** |||||||||||
> | w/o E-Notebook | 13.6 | 20.5 | 10.9 | 4.2 | 1.3 | 19.3 | 24.8 | 10.3 | 0.0 | 0.0 |
> | w/ E-Notebook | 24.9 | 34.9 | **25.0** | 5.9 | **7.5** | 35.4 | 41.5 | 29.4 | 8.8 | 7.1 |
> | w/ E-Notebook+GC | **25.9** | **37.7** | 20.5 | **11.0** | 5.0 | **36.4** | **42.6** | **29.4** | **11.8** | **7.1** |
> | **Gemini 1.5 Pro NS** |||||||||||
> | w/o E-Notebook | 29.9 | 44.3 | 21.2 | 13.6 | 6.2 | 43.0 | 52.1 | 23.5 | 17.6 | 14.3 |
> | w/ E-Notebook | 32.4 | 49.3 | 22.4 | 11.9 | 6.2 | 46.7 | 57.1 | 25.0 | **17.6** | 14.3 |
> | w/ E-Notebook+GC | **36.2** | **51.8** | **26.3** | **17.8** | **12.5** | **50.3** | **60.6** | **30.9** | 14.7 | **21.4** |
> | **Cloud Vision + Gemini 2.0 Flash** |||||||||||
> | w/o E-Notebook | 35.0 | 51.8 | 25.0 | 14.4 | 8.7 | 50.0 | 58.9 | 38.2 | 17.6 | 7.1 |
> | w/ E-Notebook | 40.4 | 59.3 | 30.1 | 16.1 | 11.3 | 57.8 | 66.3 | 47.1 | **29.4** | 7.1 |
> | w/ E-Notebook+GC | **43.2** | **63.2** | **32.7** | **17.8** | **11.3** | **62.3** | **73.0** | **48.5** | 20.6 | **14.3** |
> | **Gemini 2.5 Pro NS** |||||||||||
> | w/o E-Notebook | 36.5 | 55.1 | 25.6 | 14.4 | 6.2 | 54.0 | 65.2 | 35.3 | 20.6 | 0.0 |
> | w/ E-Notebook | 42.2 | 60.9 | **30.8** | **21.2** | **11.3** | 59.5 | 69.5 | **42.6** | 29.4 | **14.3** |
> | w/ E-Notebook+GC | **42.9** | **64.8** | 28.8 | 20.3 | 5.0 | **62.1** | **74.1** | 38.2 | **32.4** | 7.1 |
> | **Gemini 2.0 Flash NS** |||||||||||
> | w/o E-Notebook | 30.9 | 46.8 | 21.2 | 12.7 | 5.0 | 44.2 | 53.5 | 23.5 | 20.6 | 14.3 |
> | w/ E-Notebook | **40.4** | **58.2** | **31.4** | 19.5 | 8.7 | 56.8 | **67.0** | 39.7 | 23.5 | 14.3 |
> | w/ E-Notebook+GC | 40.3 | 57.3 | 30.1 | **19.5** | **13.8** | **57.0** | 66.3 | **39.7** | **29.4** | **21.4** |
>
> So **this grammar-based filtering step leads to a clear improvement**, and we will include these updated results in the final version of the paper.
>
> We appreciate your inspiring questions and hope that the new evidence addresses your concerns.
>
> Best regards,
>
> The Authors

---

> > ### Author Response · Authors · 2025-11-27
> >
> > Dear Reviewer f2Wn,
> >
> > We'd like to **further supplement** our response to the following question, as we now have **new and consolidated results**:
> >
> > > **...It would be great to see them on open-source models as well.**
> >
> > We sincerely apologize for the inconvenience caused by earlier incremental updates. To avoid confusion, **please consider this response as the single, complete answer** to ***Weaknesses #1*** and feel free to ignore our earlier partial replies to ***Weaknesses #1***.
> >
> > ---
> >
> > **[Results on Qwen2-VL-2B-Instruct]**
> >
> > *SG: 715 cases; <10: 361; 10-20: 156. HP: 398 cases; <10: 282*
> >
> > | | SG Overall | <10 | 10–20 | HP Overall | <10 |
> > |---------|------------|--------|----------|------------|--------|
> > | w/o E-Notebook | 0.8 (6) | 1.7 (6) | 0.0 (0) | 1.5 (6) | 2.1 (6) |
> > | w/ E-Notebook | 6.4 (46) | 12.5 (45) | 0.6 (1) | 10.8 (43) | 15.2 (43) |
> > | *Improvement* | +5.6 (+40) | +10.8 (+39) | +0.6 (+1) | +9.3 (+37) | +13.1 (+37) |
> > | **w/ E-Notebook + GC1** | 3.6 (26) | 7.2 (26) | 0.0 (0) | 6.0 (24) | 8.5 (24) |
> > | **w/ E-Notebook + GC2** | 6.6 (47) | 12.7 (46) | 0.6 (1) | 10.8 (43) | 15.2 (43) |
> > | **w/ E-Notebook + GC3 (cross-model)** | **8.4 (60)** | **16.6 (60)** | **0.0 (0)** | **14.6 (58)** | **20.6 (58)** |
> > | *Improvement (GC3 – w/o EN)* | +7.6 (+54) | +14.9 (+54) | +0.0 (+0) | +13.1 (+52) | +18.5 (+52) |
> >
> > From these results, we observe:
> >
> > - Even for a **2B open-source VLM** with a low baseline, our framework yields **large and meaningful gains**.
> > - With the Error Notebook and appropriate grammar-check strategies, **a 2B model can perform only marginally below GPT-4o mini** (as shown in Table 1).
> >
> > ---
> >
> > **[Results on Aya-Vision-8B]**
> >
> > For Aya-Vision-8B, we used **7× A40 GPUs (~36 hours)** followed by **3× H20 GPUs (~12 hours)**. All experimental settings remained identical to Table 1, except the model itself.
> >
> > **SG = 666 cases, HP = 370 cases**
> >
> > | | SG Overall | <10 | 10–20 | HP Overall | <10 | 10–20 |
> > |---|---|---|---|---|---|---|
> > | w/o E-Notebook | 16 (2.4%) | 16 | 0 | 14 (3.8%) | 14 | 0 |
> > | w/ E-Notebook | 54 (8.1%) | 53 | 1 | 51 (13.8%) | 50 | 1 |
> > | *Improvement* | +38 (3.4×) | +37 | +1 | +37 (3.6×) | +36 | +1 |
> >
> > These results again confirm the consistent benefits of the Error Notebook + RAG framework. Therefore, our method **generalizes across both proprietary and open-source VLMs**.
> >
> > ---
> >
> > Thank you very much for your time. We remain open to further discussion and greatly appreciate your feedback.
> >
> > Best regards,
> >
> > **The Authors**

---

### Official Review · Reviewer_vpJW · 2025-11-01

**Soundness:** 2
**Presentation:** 1
**Contribution:** 2
**Rating:** 2
**Confidence:** 4

**Summary:**

This paper considers a specific sub-problem in CAD tasks automatization, mainly part retrieval in the assembly. For this task a new approach, based on error notebooks on RAG is proposed, which can be used on top of any LLM model without fine-tuning or adaptataion. Experimental validation shows validity of the proposed approach.

**Strengths:**

1) The proposed method can be used without fine-tuning on top of any LLM model

**Weaknesses:**

1) Very limited scope of work due to selected problem
As far as I understand, the considered problem is retrieval of specific part in assembly by using its textual description. Each assembly in question consists of very limited amount of parts, less then 50. So the problem is like text retrieval from corpus of up to 50 texts. Yes, texts are very specific (CAD description of parts), but nonetheless the problem is small. This limits both the significance and impact of the work.

2) No related works or similar methods or comparision with existing methods.
How such problems are solved now? Are there any approaches for part retrieval, other then LLMs? Has anyone considered the same problem before, or it is a first time?

3) Unclear description and presentation.
Up till section 2.2 i have been struggling to understand what the problem in question is. Overall the presentation is very unclear.

**Questions:**

Please address the weakness #2 and comment on problem statement and significance/impact of your work.

---

> ### Author Response · Authors · 2025-11-20
>
> Dear Reviewer vpJW,
>
> Thank you very much for providing your constructive feedback.
>
> **Below, we respond to each of your comments point by point, and we hope our clarifications adequately address your concerns.**
>
> ***
>
> ### **[For the Weaknesses section]**
>
> > **W1: Very limited scope of work due to selected problem As far as I understand, the considered problem is retrieval of specific part in assembly by using its textual description. Each assembly in question consists of very limited amount of parts, less then 50. So the problem is like text retrieval from corpus of up to 50 texts...**
>
> W1Re: Thank you for raising this concern. We apologize that the scope was not clearly communicated. To clarify this, we have included a comprehensive illustration (**see 1_scope.pdf in the updated Supplementary Material**).
>
> On top of that, we'd like to clarify 5 points:
>
> **1. As shown in 1_scope.pdf, this task is not text retrieval. Many retrieval tasks are purely LLM-based, whereas ours is multimodal, i.e., VLM-based.** Therefore, the core problem is multimodal reasoning that bridges long, non-natural-language engineering metadata (STEP) and natural-language text data (Specification).
>
> **2. The data we process (STEP files, as shown in 1_scope.pdf) are not textual descriptions, but:**
>
> - complex, structured 3D engineering **metadata** encoding geometry, and hierarchical assembly structure;
> - whose interdependent entities are **interwoven** across the STEP file, rather than **continuous** text segments.
>
>
> **3. Significance for CAD automation**
>
> Manufacturing specifications are expressed in natural language,  while STEP files encode the parametric 3D geometry that's long and complex to interpret. Automatically linking the two is a key bottleneck in CAD automation. **Our two-stage VLM pipeline (VLM-1 for geometric interpretation, VLM-2 for NL reasoning), together with the Error Notebook/RAG, provides a practical, training-free solution for automatically interpreting STEP files.**
>
> **4. Significance for the AI community**
>
> GPT, Gemini, and other proprietary models are top-tier VLMs, yet we cannot finetune them due to the lack of model weights. Our framework:
>
> - shows how our proposed Error Notebook/RAG enhance performance,
> - significantly improves their performance **without any training**,
> - provides a reusable paradigm for domain adaptation.
>
> **5. Additional clarification:** our method supports assemblies from **2 to 249 parts** (as shown in 1_scope.pdf and Table 1), not only <50.
>
> Thank you again, and we have added this figure to the revised main paper.
>
>
> > **W2: No related works or similar methods or comparision with existing methods. How such problems are solved now? Are there any...**
>
> W2Re: Thank you for this important question.
>
> To the best of our knowledge, no prior work has addressed specification-aware **part retrieval** from long, non-natural-language STEP metadata. Since STEP metadata is highly complex for humans to interpret, VLMs are currently **the only plausible way** to process such information. **Our work is the first to bridge natural-language engineering specifications** with **STEP-based 3D part retrieval via VLM reasoning**.
>
> **It should be noted that the core of our paper is a training-free VLM methodology**. In the original submission, we **compared** models with vs without Error Notebook.
>
> Based on your comments, we further include the following **comparisons with existing methods**:
>
> - **Existing baseline comparison: standard few-shot learning**
> - **Existing baseline comparison: self-consistency**
> - **Existing open-source VLM comparison: Qwen2-VL-2B-Instruct**
>
> Below are the experimental results.
>
> ---
>
> **[Existing baseline comparison]**
>
> *SG=Self-generated dataset (715 cases), HP=Human preference dataset (398 cases)*
>
> | | SG: Overall | <10 | 10–20 | 20–50 | >50 | HP: Overall | <10 | 10–20 | 20–50 | >50 |
> |-|-|-|-|-|-|-|-|-|-|-|
> | **standard few-shot** | 26.6 | 37.4 | 19.2 | 16.9 | 6.2 | 37.7 | 42.9 | 29.4 | 17.6 | 21.4 |
> | **w/o E-Notebook** | 28.5 | 40.7 | 22.4 | 15.3 | 5.0 | 41.7 | 47.9 | 32.4 | 26.5 | 0.0 |
> | **self-consistency** | 38.9 | 54.6 | 30.1 | 21.2 | 11.3 | 54.8 | 61.7 | 42.6 | 29.4 | 35.7 |
> | **w/ E-Notebook (ours)** | **48.3** | **66.8** | **35.9** | **29.7** | **16.3** | **65.1** | **75.5** | **42.6** | **41.2** | **21.4** |
>
> **Conclusion: Our method outperforms existing baselines: standard few-shot, self-consistency, and the zero-shot w/o E-Notebook baseline across all groups.**
>
> ---
>
> **[Existing open-source VLM comparison: Qwen2-VL-2B-Instruct]**
>
> | | SG: Overall | <10 | 10–20 | HP: Overall | <10 |
> |-|-|-|-|-|-|
> | **w/o E-Notebook** | 6 | 6 | 0 | 6 | 6 |
> | **w/ E-Notebook** | 46 | 45 | 1 | 43 | 43 |
> | **Improvement** | **+40 (6.7×)** | +39 | +1 | **+37 (7.2×)** | +37 |
>
> **Conclusion: Even for a 2B model (Qwen2-VL-2B-Instruct), Error Notebook + RAG brings large and meaningful gains**.
>
> All results and clarifications have been included in the revised submission.

---

> ### Author Response · Authors · 2025-11-20
>
> > **W3: Unclear description and presentation. Up till section 2.2 i have been struggling to understand what the problem in question is. Overall the presentation is very unclear.**
>
> W3Re: Thank you very much for your feedback. sincerely apologize for the confusion! Please see 1_scope.pdf in the updated Supplementary Material. We hope this visual explanation helps clarify the problem in question.
>
> ***
>
> ### **[For the Questions section]**
>
> > **Q: Please address the weakness #2 and comment on problem statement and significance/impact of your work.**
>
> QRe: Thank you. Please firstly refer to our response to weakness #2. Then,
>
> **1. Regarding the problem statement:**
>
> The core problem is multimodal reasoning that bridges long, non-natural-language engineering metadata (STEP) and natural-language text data (Specification). Specifically, the task involves **retrieving (symbolic) text from long, non-natural-language metadata using a natural-language specification.**  **Please refer to 1_scope.pdf for a clear visualization**.
>
> **2. Regarding the significance/impact of our work:**
>
>  - We propose a training-free reasoning framework that combines the **Error Notebook and RAG** for VLM inference. The Error Notebook is constructed by systematically collecting and correcting CoT errors. **Results show that our method outperforms existing baselines: standard few-shot, self-consistency, and the zero-shot w/o E-Notebook baseline across all groups. Even for small open-source models, the method brings substantial performance improvements**.
>
>  - We reconstruct a new multimodal **human-in-the-loop dataset**. To better benefit the AI community, we provide the human-annotated dataset (see **dataset** in the updated Supplementary Material).
>
>  - We design an effective two-stage VLM strategy that overcomes the challenge of processing extremely long, non-natural-language metadata inputs.
>
> **We hope these clarifications help address the concerns and allow for a re-assessment of our contribution.**

---

> > ### Author Response · Authors · 2025-11-25
> >
> > Dear Reviewer vpJW,
> >
> >
> > Thank you again for your earlier comments!
> >
> > To further address your concerns, we have added a new **Scope of Work figure at the very beginning of the updated submission (before the Abstract)**. We sincerely hope this revised presentation helps clarify the exact task we are studying and allows you to re-evaluate our contributions.
> >
> > Below we summarize the key contributions of our work as they now appear in the updated version:
> >
> > (1) We propose a training-free reasoning framework that combines the Error Notebook with RAG for VLM inference. Importantly, **the method surpasses traditional training-free inference-time approaches (*standard few-shot*, *self-consistency*) and also demonstrates strong improvements on open-source models (e.g., *Qwen2-VL-2B-Instruct* and *Aya-Vision-8B*)**.
> >
> > (2) We construct a new CAD assembly dataset with relational specifications and human preference annotations. **The entire human-annotated dataset has been released to ensure reproducibility and facilitate future research.**
> >
> > (3) From an engineering perspective, our two-stage pipeline effectively handles the challenge of interpreting extremely long, non-natural-language STEP-file metadata.
> >
> > (4) The construction workflow of CoT correction is novel and is described in detail in the newly added **2_CoT.pdf** in the updated Supplementary Materials.
> >
> >
> > We warmly welcome any further discussion you may have. Thank you very much for your time and consideration.
> >
> >
> > Sincerely,
> >
> > The Authors

---

> > > ### Author Response · Authors · 2025-11-27
> > >
> > > Dear Reviewer vpJW,
> > >
> > > Thank you so much for your comments. To further address your question regarding:
> > >
> > > > **...significance/impact of your work**
> > >
> > > please allow us to present additional results. We have applied our Error Notebook to both proprietary models and open-source models, and found that with the Error Notebook and the necessary grammar-check strategies, **our approach enables a 2B model to perform only marginally below GPT-4o mini**. Below are the results on the open-source model:
> > >
> > > *SG: 715 cases; <10: 361; 10-20: 156. HP: 398 cases; <10: 282*
> > >
> > > | | SG Overall | <10 | 10–20 | HP Overall | <10 |
> > > |---------|------------|--------|----------|------------|--------|
> > > | w/o E-Notebook | 0.8 (6) | 1.7 (6) | 0.0 (0) | 1.5 (6) | 2.1 (6) |
> > > | w/ E-Notebook | 6.4 (46) | 12.5 (45) | 0.6 (1) | 10.8 (43) | 15.2 (43) |
> > > | *Improvement* | +5.6 (+40) | +10.8 (+39) | +0.6 (+1) | +9.3 (+37) | +13.1 (+37) |
> > > | **w/ E-Notebook + GC1** | 3.6 (26) | 7.2 (26) | 0.0 (0) | 6.0 (24) | 8.5 (24) |
> > > | **w/ E-Notebook + GC2** | 6.6 (47) | 12.7 (46) | 0.6 (1) | 10.8 (43) | 15.2 (43) |
> > > | **w/ E-Notebook + GC3 (cross-model)** | **8.4 (60)** | **16.6 (60)** | **0.0 (0)** | **14.6 (58)** | **20.6 (58)** |
> > > | *Improvement (GC3 – w/o EN)* | +7.6 (+54) | +14.9 (+54) | +0.0 (+0) | +13.1 (+52) | +18.5 (+52) |
> > >
> > > Under the **cross-model distillation** setting, the model achieves the best results:
> > >
> > > - **On the human-preference dataset (<10 group), this 2B model is only 4.2 points below the proprietary GPT-4o mini (as shown in Table 1).**
> > > - The absolute overall accuracy of this 2B model on HP is even *higher* than GPT-4o mini on SG.
> > >
> > > We hope this result may help you reevaluate our work. If you have any additional questions, we would be very happy and open to continue the discussion! Thank you.
> > >
> > > Sincerely,
> > > **The Authors**

---

### Author Response · Authors · 2025-11-20

Dear Reviewers,

Thank you very much for your valuable feedback and constructive suggestions. Although revising our work and conducting additional experiments has been challenging, your insights have genuinely motivated us and significantly strengthened the paper.

**We kindly invite you to review our detailed responses for each reviewer.**

In addition, we have updated:

1. **The main paper**
2. **The supplementary materials**, now including:
   - our implementation code,
   - the human-annotated dataset,
   - a *scope illustration PDF* that clarifies the problem setting.
   - an *illustration PDF* that clarifies the verification of CoT traces.

We sincerely hope that these improvements will help you re-evaluate our work.

Thank you again for your time, careful consideration, and continued support.

Best regards,
The Authors

---

### Author Response · Authors · 2025-11-25

Dear AC and Reviewers,

Thank you very much for your valuable feedback, constructive suggestions, and support throughout the review process. Although conducting additional experiments has been challenging, **your input has truly motivated us and helped strengthen our paper**. Here, we summarize all our proposed changes (**as candidates, and we welcome any further guidance or requirements from the AC and Reviewers**):

***

**Notation**:

- *main*: main paper
- *supp.*: supplementary material
- *extra*: additional 1-page extension allowed by ICLR 2026 for accepted papers
- ~: approximately (e.g., Line ~20 = around line 20; final version may shift slightly)

***

**[1] Changes we have made that clarify experiments already present in the paper**

- **Line ~10, main**: Add **a scope-of-work figure** to clearly illustrate the task.
- **Line ~52, main**: Add: **our method surpasses traditional training-free inference-time approaches (standard few-shot, self-consistency) and further demonstrates strong improvements even on open-source models (e.g., Qwen2-VL-2B-Instruct, Aya-Vision-8B). Under the cross-model GC setting, where the Error Notebook is constructed using GPT-4o (Omni), the 2B model inference achieves performance that comes within roughly 4 points of GPT-4o mini.**
- **Line ~198, main**: Add missing comma and equation number.
- **Line ~207, main**: Add missing comma and equation number.
- **Line ~230, main**: Add missing comma and equation number.
- **Line ~235, main**: Add equation number.
- **Line ~247, main**: Add a new subsection and the description of **grammar constraint (GC) mechanisms** (w/ updated definitions of GC1 / GC2 / GC3).
- **Line ~299, main**: Add missing comma and equation number.
- **Line ~314, main**: Add missing comma and equation number.
- **Line ~1062, supp.**: Add the overview figure: verification of the corrected reasoning trajectories / CoT traces.

***

**[2] Changes that are entirely new**

- **Line ~378, main**: Add the GC experiments for proprietary models, Table 1.
- **Line ~454, main**: Add comparison with existing baselines: **standard few-shot** and **self-consistency**.
- **Line ~486, extra**: Add comparisons with the open-source VLMs, including **Qwen2-VL-2B-Instruct** and **Aya-Vision-8B**.
- **Line ~496, extra**: Add results under the grammar constraint setting.
- **Line ~501, extra**: Add **token usage**, **latency overhead**, and **API cost** analysis.
- **Line ~924, supp.**: Add evaluation metrics for **retrieval relevance**.
- **Line ~948, supp.**: Add explanation of the **retrieval function** used for RAG.

***

**Thank you again for your careful consideration and support!**

Best regards,

The Authors

---

### Author Response · Authors · 2025-11-26

Dear AC and Reviewers,

We sincerely appreciate the constructive feedback and encouragement. Below we summarize the key points of the current paper so far.

---

### **(1) Significance of our work**

This paper presents the **first training-free inference-time adaptation framework (Error Notebook + RAG)** for **specification-aware CAD part retrieval**, supported by a **new human-annotated multimodal CAD dataset** and a **two-stage VLM pipeline** capable of handling long STEP metadata.

(1) We propose a training-free reasoning framework combining the *Error Notebook* and RAG. In constructing the Error Notebook, **corrected CoTs are linked through reflective corrections until the correct solutions are obtained**.

(2) We introduce a grammar-constraint (GC) verifier to ensure the structural validity of corrected CoTs used in the **Error Notebook**.

(3) We reconstruct a multimodal CAD assembly dataset with relational specifications and human preference annotations. **We have fully released the code and human-annotated dataset for reproducibility.**

(4) We design a two-stage VLM strategy that first generates part descriptions and then performs retrieval, effectively addressing the challenge of processing long, non-natural-language STEP files.

---

### **(2) Significant results**

- Under cross-model grammar-check, using the Error Notebook, **the Qwen2-VL-2B-Instruct model achieves performance that comes within roughly 4 points of GPT-4o mini**.
- Across proprietary VLMs (GPT-4o Omni, GPT-4o mini, Gemini 2.5 Pro, Gemini 2.0 Flash, Gemini 1.5 Pro, etc.), our method yields substantial gains, with GPT-4o Omni reaching **+23.4% absolute accuracy** on the human-preference dataset.
- Our method **surpasses traditional training-free approaches**: standard few-shot, self-consistency.
- It also shows **strong improvements on open-source models**, including **Qwen2-VL-2B-Instruct** and **Aya-Vision-8B**.
- The Error Notebook is especially beneficial for **challenging high-part-count assemblies (>10)**.
- **Token usage, latency, and API-call analysis** demonstrate that the method is efficient.
- **Retrieval relevance** results confirm that the Error Notebook improves retrieval quality as well.

---

We truly hope these updates help re-evaluate our contribution.
Thank you very much for your time and careful consideration.

Best regards,
The Authors

---

### Meta-Review · Area_Chair_iEyb · 2025-12-27

**Summary:**

The paper proposes an inference-time adaptation framework for specification-aware CAD part retrieval. The key idea is to construct an Error Notebook consisting of corrected chains-of-thought (CoTs), which are then retrieved via RAG to guide inference. The approach is training-free and demonstrates substantial accuracy improvements over standalone VLM baselines across multiple models and datasets.

Overall, most reviewers agree that the idea is interesting and practically useful, and the empirical results are consistently strong, showing clear and significant improvements over relevant training-free baselines. While limitations remain and the numerical scores are mixed, the method demonstrates clear advantages over existing baselines. Considering both the reviews and the rebuttal, I am leaning toward a positive recommendation.

**Reviewer Concerns:**

At this stage, the remaining concerns can be summarized as follows:
1- the lack of comparison with traditional CAD retrieval methods,
2- the reliance on ground-truth labels to construct the Error Notebook, which limits scalability,
3- the absence of a strong theoretical justification despite strong empirical results, and
4- questions regarding the scope and broader impact of the task.

**Reviewer Scores:**

Based on the scores alone, the reviews lean slightly negative. However, a close reading of the reviews and rebuttal suggests that several concerns have been addressed by the authors during the discussion phase, including over-claiming, evaluation on open-source models, retrieval relevance metrics, and efficiency analysis. In particular, at least one reviewer explicitly raised their score after the rebuttal, and another maintained a positive assessment. It is likely that additional reviewer engagement could further improve the overall evaluation.

---

### Decision · Program_Chairs · 2026-01-26

Accept (Poster)